# High-resolution mapping of monthly industrial water withdrawal in China from 1965 to 2020

Chengcheng Hou[1,2], Yan Li[1,2, *], Shan Sang[1,2], Xu Zhao[3], Yanxu Liu[1,2], Yinglu Liu[4,5], Fang Zhao[6]

[1] State Key Laboratory of Earth Surface Processes and Resources Ecology, Beijing Normal University, Beijing, 100875, China
[2] Institute of Land Surface System and Sustainable Development, Faculty of Geographical Science, Beijing Normal University, Beijing, 100875, China
[3]Institute of Blue and Green Development, Shandong University, Weihai, 264209, China
[4]College of Urban and Environmental Sciences, Peking University, Beijing, 100871, China
[5]Key Laboratory for Earth Surface Processes of the Ministry of Education, Peking University, Beijing, 100871, China
[6]Key Laboratory of Geographic Information Science of the Ministry of Education, School of Geographic Sciences, East China Normal University, Shanghai 200241, China.

*Correspondence to*: Yan Li (yanli.geo@gmail.com)

**Abstract.** High-quality gridded data on industrial water use are vital for research and water resource management. However, such data in China usually have low accuracy. In this study, we developed a gridded dataset of monthly industrial water withdrawal (IWW) for China, which is called the China industrial water withdrawal dataset (CIWW); this dataset spans a 56-year period from 1965 to 2020 at spatial resolutions of 0.1° and 0.25°. We utilized >400,000 records of industrial enterprises, monthly industrial product output data, and continuous statistical IWW records from 1965 to 2020 to facilitate spatial scaling, seasonal allocation, and long-term temporal coverage in developing the dataset. Our CIWW dataset was significantly improved in comparison to previous data for the characterization of the spatial and seasonal patterns of the IWW dynamics in China and showed consistency with statistical records at the local scale. The CIWW dataset, together with its methodology and auxiliary data, will be useful for water resource management and hydrological models. This new dataset is now available at https://doi.org/10.6084/m9.figshare.21901074 (Hou and Li, 2023).

## 1 Introduction

Industrial water withdrawal (IWW) is the amount of water abstracted from freshwater sources for industrial purposes, which is different from water consumption; IWW accounts for approximately 19% of human water withdrawal globally and is the second largest sector of human water use following irrigation (WWAP, 2019). In developed countries, IWW accounts for more than half of their water use (Shen et al., 2010; Wada et al., 2011a; Flörke et al., 2013). Driven by economic and population growth, global IWW has steadily increased over the past 60 years (Oki and Kanae, 2006; Wada et al., 2011b) from 400 km$^3$ per year in 1960 to 955 km$^3$ per year in 2010 (Flörke et al., 2013), and it is projected to continue to increase in the future (Oki et al., 2003; Shen et al., 2010; Fujimori et al., 2017). Considering the high spatial heterogeneity and fast changes in IWW, quantitative information with high spatiotemporal resolution on IWW is essential for water resource management and research.

Existing IWW data primarily consist of statistical data at the administrative/watershed levels and model estimations at the grid level, in which the sectoral information is represented with varying degrees of complexity (Arnell, 1999, 2004; Alcamo et al., 2000, 2007; Vörösmarty et al., 2000; Oki et al., 2003; Hanasaki et al., 2008a; Otaki et al., 2008; Wada et al., 2011b; Hejazi et al., 2014; Wada et al., 2016; Yan et al., 2022). Gridded datasets developed from administrative-level data or models provide more detailed spatial information (Hanasaki et al., 2008a; Wada et al., 2011a); however, their accuracy depends on the downscaling methods, including the spatial proxies and data sources.

For the total IWW, statistical data are usually allocated to the grid level relying on the spatial proxies, such as population density and urban or industrial area (Hanasaki et al., 2008a, b, 2010; Van Beek et al., 2011; Wada et al., 2011a, b, 2014). For sectoral IWW, different mapping methods are applied. For the energy sector, water withdrawal was estimated by the total energy generated and water use efficiency under different technologies (Koch and Vögele, 2009; Flörke et al., 2013). With detailed information on the location, power output, and water use efficiency of power plants, water withdrawal for the energy sector could be mapped out (Vassolo and Döll, 2005; Flörke et al., 2013; Müller Schmied et al., 2014; Wang et al., 2016; Qin et al., 2019). For manufacturing, water withdrawal was estimated either as the residue of the energy water use from the total IWW downscaled using the spatial proxies mentioned above (Hejazi et al., 2014) or the product of population and per capita water consumption (Vörösmarty et al., 2000). Although several global gridded IWW datasets have been developed, the spatial proxies used for downscaling (e.g., population) are only indirect factors that are not directly tied to industrial production processes that consume water, and they cannot be used to separate the different industrial subsectors whose water use efficiency could be substantially different (0.32 of Paper and Paper Products versus 5.6 of Electric Equipment and Machinery, unit: $10^3$ yuan/m$^3$). Moreover, when downscaling, the global gridded datasets typically rely on the national statistical data (Hejazi et al., 2014; Water GAP model 2.2 (Wada et al., 2016); Huang et al., 2018) without incorporating subnational statistics to better capture the regional differences. Therefore, global datasets are sufficient in showing the global general pattern but can have poor performance for the specific regions, limiting their applications for regional water issues (Liu et al., 2019b).

IWW has seasonal fluctuations because of changes in weather conditions (temperature, precipitation, and thunderstorms), water supply availability (especially under monsoon climates, such as in China), production demand, and emission restrictions (Liu et al., 2006). However, most existing datasets do not represent seasonal variations (only annual data) or simply use monthly invariable estimation (i.e., each month shares 1/12 of annual total withdrawal) (Brunner et al., 2019; Wada et al., 2011a). The lack of representation of intra-annual variations may result in significant discrepancies between the data and reality. A few studies consider seasonal variations in water withdrawal for specific sectors. For example, seasonality in IWW for electricity generation is estimated by incorporating the influence of temperature variability on the cooling water demand of thermoelectric power plants (Byers et al., 2014; Liu et al., 2015). The included climate variations introduce a clear seasonal pattern, with large withdrawals in winter at high latitudes and summer in tropical regions (Huang et al., 2018). Therefore, it is essential to fully account for intra-annual variations in IWW, which directly affect water resource management and allocation (Derepasko et al., 2021; Sunkara and Singh, 2022).

After decades of fast growth, China has become the second-largest economy in the world, with rapid industrial development leading to increasing water use (Zhou et al., 2020). IWW in China accounted for 20.2% of the total water withdrawal in 2019 (source: China Water Resources Bulletin) and increased by 4.5 times from 31.93 km$^3$ in 1965 to 142.86 km$^3$ in 2013 (Zhou et al., 2020). However, water resources in China are distributed unevenly in space, causing severe water stress due to a mismatch between the water supply and demand of the population and industrial development (Liu et al., 2013; Zhao et al., 2015). For

instance, Northern China is one of China's largest industrial centres and densely populated regions, but it is experiencing the most severe water scarcity in the world (Yin et al., 2020). The changes in IWW and total water withdrawal have further increased the water conflict, making it urgent to optimize the current water use and management structure. Therefore, high-quality gridded IWW data for China are needed to characterize the spatial-temporal pattern of IWW for water management and for research on hydrological processes and modelling (Addor et al., 2020). However, IWW data produced from reliable

data sources with a long period and high spatial resolution in China are still lacking. The publicly available data on IWW in China are either the statistical data at the provincial, prefecture, or basin level (Xia et al., 2017; Qin et al., 2020; Chen et al., 2021) or the gridded data extracted from the global datasets that have low accuracy for regional and local studies (Liu et al., 2019a, b; Han et al., 2019; Niva et al., 2020; Yin et al., 2020; Li et al., 2022).

To address this data gap, in our study, we used reliable local data sources to develop gridded datasets of monthly IWW in

China with high spatial resolution and seasonal variations. By using multiple statistical data, the high-resolution mapping of IWW was achieved by a unique industrial enterprise dataset including >400,000 enterprises; the seasonal variations were derived from the industry product output data; and the long-term temporal coverage was obtained by the continuous statistical records from 1965 to 2020. The resulting dataset, named the China Industrial Water Withdrawal dataset (CIWW), provides monthly IWW from 1965 to 2020 at spatial resolutions of 0.1° and 0.25°. The dataset is useful to better understand the spatial

and seasonal variations in IWW in China and support hydrological studies and regional water resource management.

## 2 Data and Method

### 2.1 Data

#### 2.1.1 Statistical data for industrial output value and water withdrawal

The provincial-level industrial output value (IOV, unit: 10$^3$ Yuan per year) and IWW were from the China Economic Census

Yearbook in 2008 (http://www.stats.gov.cn/sj/pcsj/jjpc/2jp/indexch.htm, last accessed: 2 April 2021). The data included surveyed IOV and IWW for enterprises above a designated production level, consisting of three main industrial sectors (mining, manufacturing, and production and supply of electricity, gas and water) and 38 subsectors (Table A1). Note that the two subsectors of "Other Mining" and "Waste Resources and Material Recycling and Processing" contained no data, and the average values of the IOV and IWW of the mining and manufacturing sector in each province were used to fill these two

subsectors.

### 2.1.2 Industrial enterprise data in China

The industrial enterprise dataset used in this study was from the Chinese Industrial Enterprise Database in Mainland China from 1998 to 2013 (https://www.lib.pku.edu.cn/portal/cn/news/0000001637, last accessed: 18 May 2022). The dataset contains surveyed industrial information, including address, products, annual IOV, and industry category, for more than 400,000 enterprises whose annual IOV was more than 5 million Yuan (or 20 million Yuan from 2011 to 2013 due to standard changes). The dataset covers three main industrial sectors and 37 subsectors, similar to the provincial data in Section 2.1.1. The enterprises' records for the subsector of "Water Production and Supply" were not used because the water supply was mainly for domestic rather than industrial purposes. To match the IWW survey data, which were only available in 2008 (the economic censuses in other years do not include detailed provincial IWW by subsector), industrial enterprise data in 2008 were selected for spatial downscaling of the provincial IWW (Fig. B2).

### 2.1.3 Statistical data for the monthly industrial product output

The monthly industrial product output data were from the China Industry Product Output Database (http://olap.epsnet.com.cn, last accessed: 26 September 2021). The data contain monthly outputs of 283 specific products of 36 industrial subsectors at the provincial level. We used the average of 5 years from 2006-2010 to reduce interannual variability in outputs. The monthly outputs of each product were converted to monthly fractions (divided by the annual total output) to represent its intra-annual variation. Missing values in monthly product output fractions were filled by the average value of monthly fractions of product output from 2006 to 2010. The monthly output fractions of 283 products were aggregated to 36 subsectors by averaging products within each subsector by the arithmetic mean.

### 2.1.4 Statistical data of the industrial water withdrawal for long-term extension

Long-term statistical IWW data were required to produce IWW data for the past four decades. Provincial surveyed statistical data on IWW in China from the China Water Resources Bulletin (http://www.mwr.gov.cn/sj/tjgb/szygb/, last accessed: 3 May 2022) from 2003 to 2020 were used. IWW in the China Water Resources Bulletin is defined as the annual amount of water withdrawal for industrial production activities, including primary production, auxiliary production and ancillary production, excluding recycled water. To further extend the time series to an earlier period, the IWW reported by Zhou et al., (2020) (referred to as 'Zhou2020 data' hereafter) from 1965 to 2002, was used after summing the prefecture data to the provincial level; its IWW was from multiple versions of water resources survey data (1st and 2nd National Water Resources Assessment Program) and defined the same way as the China Water Resource Bulletin and our study. The national IWW between two sources (Zhou2020 data and China Water Resources Bulletin) was almost identical in 2003 (117.72 vs 118.86 unit: km$^3$; Fig. C3) but started to diverge afterward. To ensure data continuity, we opted for the China Water Resources Bulletin starting from 2003 as a statistical data source because it has been updated continuously since then. Thus, the combination of the above two

data sources provided complete and continuous statistical records of IWW from 1965 to 2020 in China. Table 1 provides a summary of the data sources used for developing the CIWW dataset.

**Table 1 A summary of data sources for developing the CIWW dataset**

| Data | Source | Industrial Sector | Spatial resolution | Time span | Purpose |
|---|---|---|---|---|---|
| Industrial enterprise output value | Chinese Industrial Enterprises Database | | Point | Yearly, 2008 | |
| Industrial water withdrawal | China Economic Census Yearbook | Subsectors (36) | Province | Yearly, 2008 | Spatial mapping |
| Industrial output value | | | | | |
| Monthly product output (283 products) | China Industry Product Output Database | | Province | Monthly, 2006-2010 | Seasonal allocation |
| Industrial water use | China Water Resources Bulletin | None | Province | Yearly, 2003-2020 | Long-term data from 1965 to 2020 |
| Industrial water use | Zhou et al., 2020 | Sectors (10) | Prefecture | Yearly, 1965-2002 | |

### 2.1.5 Other industrial water withdrawal data for comparison

There are two other gridded IWW datasets used to compare with the CIWW dataset: the global gridded monthly sectoral water use dataset for 1971-2010 at 0.5° (Huang et al., 2018) (referred to as 'Huang data' hereafter) and water abstraction for industrial uses from 1901 to 2005 at 0.5° as the input data for ISIMIP2b (referred to as 'model data' hereafter). The IWW from Huang data consists of three sectors: mining, manufacturing, and cooling of thermal power plants, and the sum of the three sectors was treated as the total IWW. The IWW from model data is the multi-model mean (Water GAP, PCR-GLOBWB, and H08).

The sum of sectoral IWW (if available) was treated as total IWW (Wada et al., 2016). Unit of IWW was converted from $m^3$ to mm by dividing the grid cell area. Table D4 provides a summary of the data description used for comparison.

### 2.2 Method

The development of the CIWW dataset primarily consisted of three steps: 1) mapping the provincial IWW data to the grid scale, 2) allocating annual IWW data to the monthly scale, and 3) producing long time series of IWW (Fig. 1).

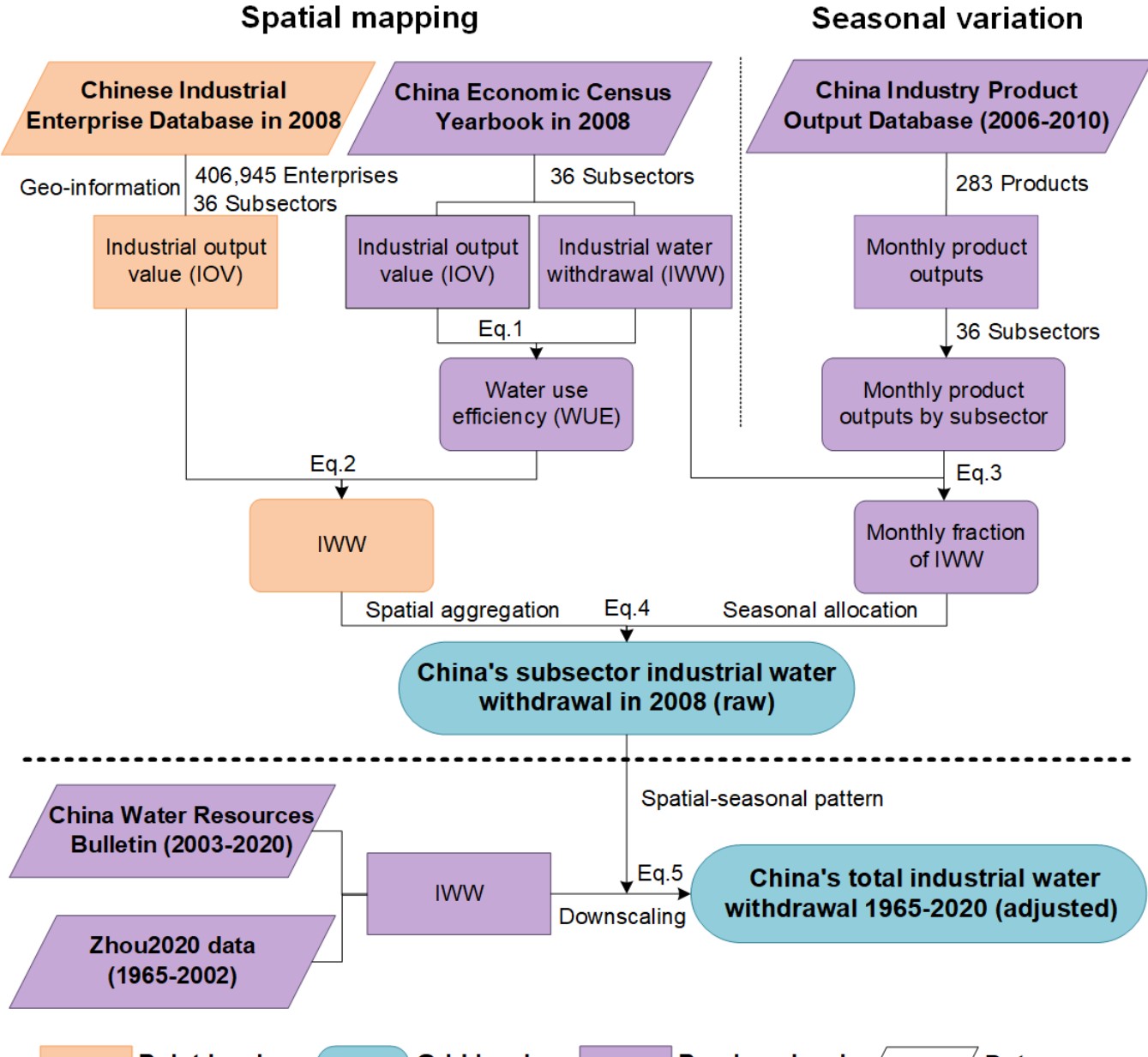

**Figure 1: Workflow for developing the CIWW dataset.**

### 2.2.1 Mapping industrial water withdrawal

The spatial mapping of IWW in China was achieved using the IOV of >400,000 enterprises in 2008 and the subsectoral water use efficiency at the provincial level from the Chinese Economic Census Yearbook in 2008.

The geographical location of industrial enterprises was obtained by converting their addresses to geographical coordinates by the BaiduV3 geocoding service with the *geopy* package in Python. The industrial water use efficiency ($WUE_{p,subs}$) of the province $p$ and subsectors *subs* was computed as the industrial output value ($IOV_{p,subs}$) divided by industrial water withdrawal ($IWW_{p,subs}$) (Eq. 1):

$$WUE_{p,subs} = \frac{IOV_{p,subs}}{IWW_{p,subs}} \tag{1}$$

By assuming a same industrial water use efficiency for all industrial enterprises in a province and the a subsector, the industrial water withdrawal ($IWW_{i,subs}$) of enterprise $i$ belonging to the subsector *subs* was estimated by multiplying the corresponding water use efficiency of the subsector *subs* in province $p$ ($WUE_{p,subs}$) and the industrial output value of enterprise $i$ ($IOV_{i,subs}$), as shown in Eq. 2:

$$IWW_{i,subs} = WUE_{p,subs} \times IOV_{i,subs} \tag{2}$$

The IWW of enterprises of specific subsectors ($IWW_{i,subs}$) could be summed up from the point level to the grid level at a given spatial resolution ($IWW_{gird,subs}$). The summation of the subsectors ($\sum_{subs=1}^{36} IWW_{grid,subs}$) provided the spatial pattern of the total IWW in 2008.

### 2.2.2 Allocating industrial water withdrawal to seasonal variations

We assumed that monthly IWW was proportional to the industrial product output and that there was no seasonal variation in water use efficiency during the year. Therefore, seasonal variations in IWW could be approximated by the monthly industrial product output, which was calculated as the monthly fractions of the product output to annual total output. The seasonal pattern included signals of variations in climate and weather because the industrial product output for some sectors could be affected by seasonal climate conditions and extreme weather events (e.g., production shutdowns or restrictions due to heatwaves, thunderstorms, torrential rains). Since the climate change-induced seasonality changes were slow and gradual, their influences on monthly IWW were also low, and the long-term climate change impacts (e.g., warming) could be captured by the yearly statistical IWW data.

Since the monthly industrial product output data included 283 different products of different subsectors and the number of products varied across subsectors, we initially calculated the monthly fraction of each product output of each province, averaged from 2006 to 2010, to reduce the influence of interannual variability. Because industrial water for producing different products is unknown, we simply used the arithmetic mean of the monthly fractions of the different products belonging to a subsector to represent aggregated monthly fractions for the subsector. In this way, we obtained the fractions of the product outputs for subsector *subs* in province $p$ for month *mon* ($Fraction_{p,mon,subs}^{output}$).

Although provincial differences exist in the seasonality of IWW, we found that $Fraction_{mon,p,subs}^{output}$ in certain subsectors and provinces exhibited unreasonable seasonal variations that were difficult to explain (Fig. E5). Instead of directly using the provincial-specific seasonal variations in output, the seasonal variations in each industrial subsector ($Fraction_{mon,subs}^{water}$) were

represented by the weighted mean of monthly product fractions across all provinces ($\text{Fraction}_{mon,p,subs}^{output}$) with weights of provincial subsector IWW ($IWW_{p,subs}$) from the Chinese Economic Census Yearbook in 2008 (Eq. 3). The only exception is for Electricity and Heating Power Production and Supply (EPS) subsector because its seasonality is strongly linked to seasonal

temperature variation of each province and thus may exhibit regional differences. To account for this issue, we used the K-means method and classified the seasonality of EPS into three types, which broadly correspond to North China (type 1), South and Northwest China (type 2), and Xizang (type 3), respectively (Fig. F6). In particular, Shanghai was manually adjusted from the originally classified type 1 to type 2 because of its strong peak in JJA.

$$\text{Fraction}_{mon,subs}^{water} = \frac{\sum_{p=1}^{31}\left(\text{Fraction}_{p,mon,subs}^{output} \times IWW_{p,subs}\right)}{\sum_{p=1}^{31} IWW_{p,subs}} \quad (3)$$

Therefore, the monthly IWW of the different subsectors at the grid level ($IWW_{grid,mon,subs}$) could be obtained by allocating its annual IWW ($IWW_{grid,subs}$) into 12 months based on the corresponding monthly fractions of the same subsector ($\text{Fraction}_{mon,subs}^{water}$) as Eq. 4.

$$IWW_{grid,mon,subs} = IWW_{grid,subs} \times \text{Fraction}_{mon,subs}^{water} \quad (4)$$

The monthly IWW at the grid level ($IWW_{grid,mon}$) after summing subsectors ($\sum_{subs=1}^{36} IWW_{grid,mon,subs}$) provided the spatial

and seasonal pattern of the total IWW of China in 2008.

**2.2.3 Developing China's industrial water withdrawal data from 1965 to 2020**

We developed long-term IWW data in China from 1965 to 2020 by mapping provincial IWW statistics based on the spatial-seasonal pattern derived from IWW in 2008. Due to the different statistical calibres of the data sources, the raw IWW from the 2008 Chinese Economic Census Yearbook was not directly used in the long-term IWW data. Instead, its spatial-seasonal

distribution was used to map the provincial industrial water withdrawal ($IWW_p$) from the China Water Resources Bulletin between 2003 and 2020 and the Zhou2020 data between 1965 and 2002. Since the Zhou2020 data showed good consistency with the China Water Resources Bulletin data, these two IWW records were combined to develop the long-term data. The provincial industrial water withdrawal ($IWW_p$) of each year was allocated to the grid level following Eq. 5 to obtain the gridded IWW data from 1965 to 2020 ($IWW_{grid,mon}^{adjust}$):

$$IWW_{grid,mon}^{adjust} = IWW_p \times \frac{IWW_{grid,mon}^{raw}}{\sum_p \sum_{mon=1}^{12} IWW_{grid,mon}^{raw}} \quad (5)$$

where $IWW_{grid,mon}^{adjust}$ was the adjusted IWW (to match $IWW_p$) of month *mon* at the grid level, $IWW_{grid,mon}^{raw}$ was the monthly IWW at the grid level in 2008, and $\sum_p \sum_{mon=1}^{12} IWW_{grid,mon}^{raw}$ summed the monthly gridded $IWW_{grid,mon}^{raw}$ to the annual total IWW of all grids in province *p*, representing the fraction of grid to provincial total IWW.

**Table 2** provides an overview of the CIWW dataset, including the gridded monthly IWW data in China from January 1965 to

December 2020 with spatial resolutions of 0.1° and 0.25° and auxiliary data supporting the development.

**Table 2 Overview of the China Industrial Water Withdrawal (CIWW) Dataset (available at https://doi.org/10.6084/m9.figshare.21901074)**

| Data | Variable | Spatial resolution | Temporal coverage | Industrial sectors |
|---|---|---|---|---|
| Main data | Industrial water withdrawal (adjusted) | 0.1°/0.25° | Monthly, 1965-2020 | NA |
| Auxiliary data | Industrial water withdrawal (raw) | 0.1°/0.25° | Monthly, 2008 | 36 subsectors |
| | Industrial output value | 0.1°/0.25° | Yearly, 2008 | 36 subsectors |
| | Number of industrial enterprises | 0.1°/0.25° | Yearly, 2008 | 36 subsectors |

## 2.3 Data validation and comparison with other datasets

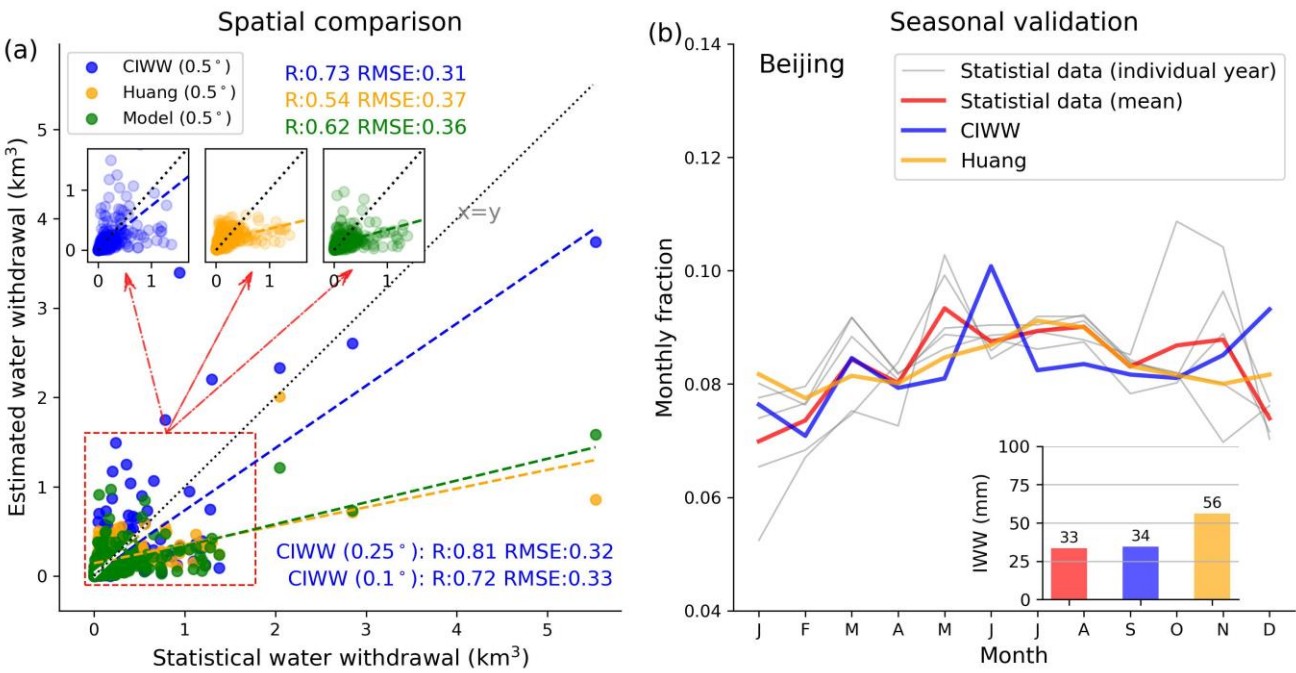

Figure 2: Validation of the CIWW data against the statistical data for spatial distribution and seasonal variation. (a) Relationship between the mean IWW of 1971-2005 from Zhou2020 data (Zhou et al., 2020) and CIWW, Huang data (Huang et al., 2018), and the model data (ISIMIP2b Input Data) for 329 prefectures in China. The black dotted line indicates the 1:1 line, and the coloured dashed lines indicate the fitted lines. For this comparison, CIWW is processed to the same spatial resolution of Huang2018 data and model data at 0.5° before aggregating to the prefecture level. Comparison results with CIWW at other resolutions (0.25° and 0.1°) are reported in R and RMSE. (b) Comparison of the 5-year mean (2006-2010) monthly variation in IWW from the surveyed data (red, (Long et al., 2020)), CIWW (blue), and Huang2018 data (green) in Beijing. The solid grey line shows IWW for individual years from 2006 to 2010. The inset shows the annual mean total IWW from 2006 to 2010.

To validate the CIWW dataset, we compared the spatial and seasonal patterns with statistical data records and other datasets.

For spatial validation, the 35-year mean IWW (1971-2005) from CIWW, global gridded data (Huang et al., 2018), and model

data (ISIMIP2b Input Data) were compared with the Zhou2020 data (treated as "truth") (Zhou et al., 2020) for 329 prefectures in China. Although we used the Zhou2020 data at the provincial level to produce the CIWW dataset, the prefecture-level data were unused in developing CIWW but left intentionally only for validation purposes. The provincial- and prefectural-level IWW are not completely independent (each province consists of many prefectures), however, the intra-provincial variations reflected in prefectural IWW are not captured by the provincial IWW. In the absence of additional validation data, the

prefectural IWW can support the validation and determine how the effectiveness of spatial patterns after downscaling. All gridded data were averaged over each prefecture using the *rasterstats* package in Python and then multiplied by the prefecture area to obtain IWW for each prefecture (in units of $km^3$). The results in Fig. 2a indicated a superior performance of CIWW data in representing the spatial variations in IWW compared against Huang data and model data due to its much higher correlation (0.73, 0.54, and 0.62) and lower root mean square error (RMSE) (0.31 vs. 0.37 vs. 0.36 km³). Additionally, when

comparing CIWW at higher resolutions (0.25°), the consistency with the Zhou2020 data improved further with higher correlation (0.81) than the 0.5° data. This result demonstrated the benefit of increased spatial resolution in characterizing the IWW at smaller scales.

    For seasonal validation, owing to the data limitation, we only had monthly surveyed statistical IWW data in Beijing from 2006 to 2010 (Long et al., 2020). The results showed that both the CIWW and Huang data could capture the 5-year mean seasonality

of IWW in Beijing (Fig. 2b). However, the magnitude of IWW was significantly overestimated by the Huang data (56 mm per year) relative to the surveyed statistical data (33 mm per year). In comparison, the magnitude of IWW in the CIWW data (34 mm per year) was more in line with the surveyed statistical data (Fig. 2b). The slight deviation of CIWW from statistical data in certain months (e.g., December) reflects the imperfect capability of applying national seasonality to characterize local variations in Beijing. These validations demonstrated better performances of CIWW data with much higher accuracy and

improved representations of the spatial and seasonal variations; thus, CIWW was a better data source for IWW-related applications in China.

## 3 Results

### 3.1 Spatial distribution of industrial water withdrawal in China

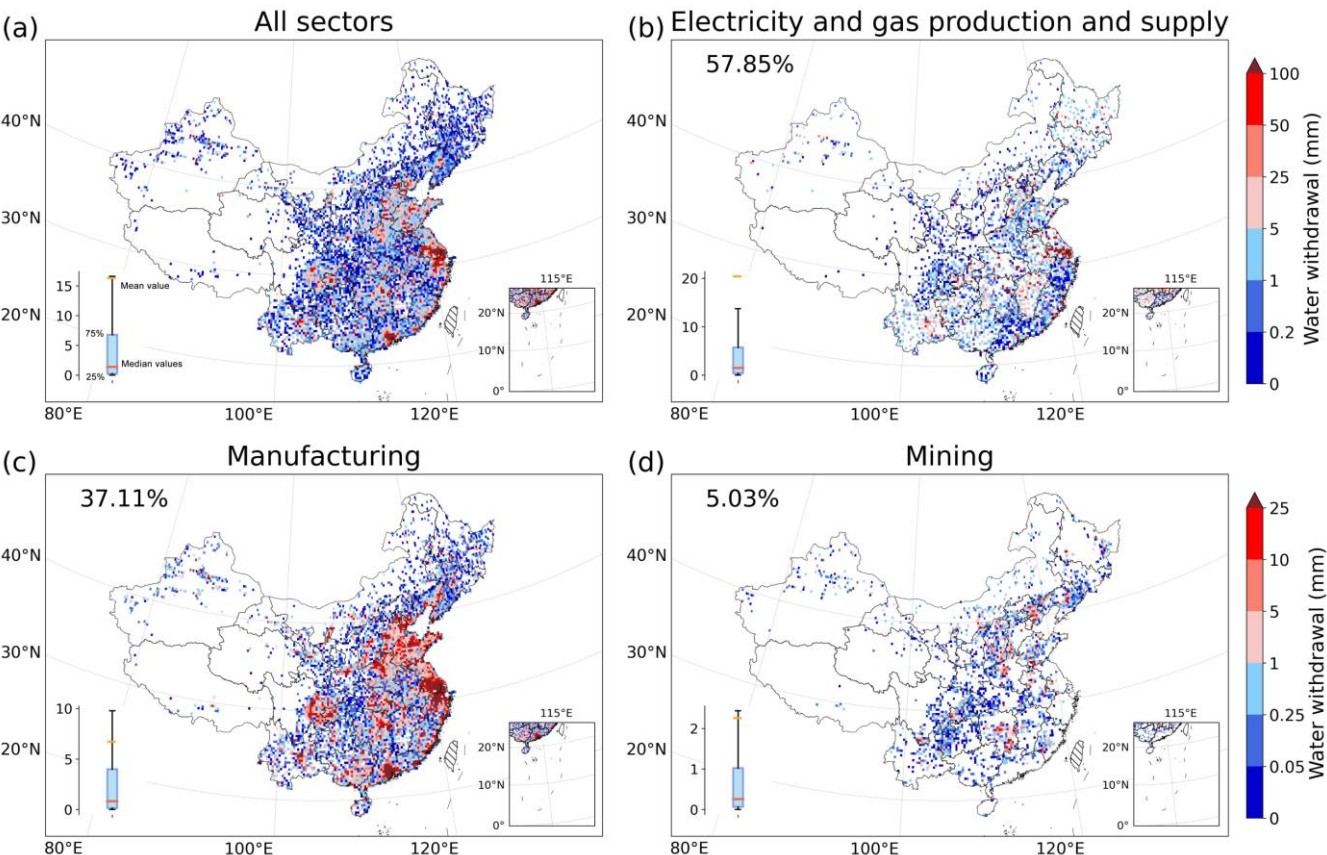

Figure 3: Total IWW (raw) in China in 2008 (a) and for different industrial sectors, including electricity and gas production and supply (EGPS, b), manufacturing (c), and mining (d). The box plot in the bottom left corner shows the interquartile range (25% and 75%) of nonzero water withdrawal, with the red and yellow lines denoting the median and mean values, respectively. The numbers displayed as percentages denote the percentage of the sectoral IWW to the total IWW.

There was substantial spatial variation in the total IWW according to the 2008 data (Fig. 3a). The eastern coastal area of China had generally higher IWW, followed by southeastern and central China, and the lowest IWW occurred in western China. The largest water withdrawal was found in the urban agglomeration of the Yangtze River Delta and Pearl River Delta. The spatial distribution of IWW over the country indicated that industry enterprises were primarily concentrated in urban areas with more intensified economic activities.

The water withdrawal by the main industrial sectors showed distinctive spatial patterns. Water withdrawal from EGPS showed a dispersive pattern that was mainly concentrated in southeastern coastal areas, especially in the Yangtze River Delta region (Fig. 3b). Water withdrawal from manufacturing broadly reflected the total IWW and population distribution of China, mainly showing the close linkage between manufacturing and population (Fig. 3c and Fig. G7). The water withdrawal of mining was

confined to regions with rich mineral resources, such as central, northern, and southwestern China (Fig. 3d). Overall, the industrial sector with the largest IWW was EGPS (57.85%), followed by manufacturing (37.11%) and mining (5.03%). The
dominance of the EGPS sector in total IWW reflected the large water requirement for thermoelectric power generation (Gu et al., 2016; Niva et al., 2020).

## 3.2 Seasonal variations in industrial water withdrawal in China

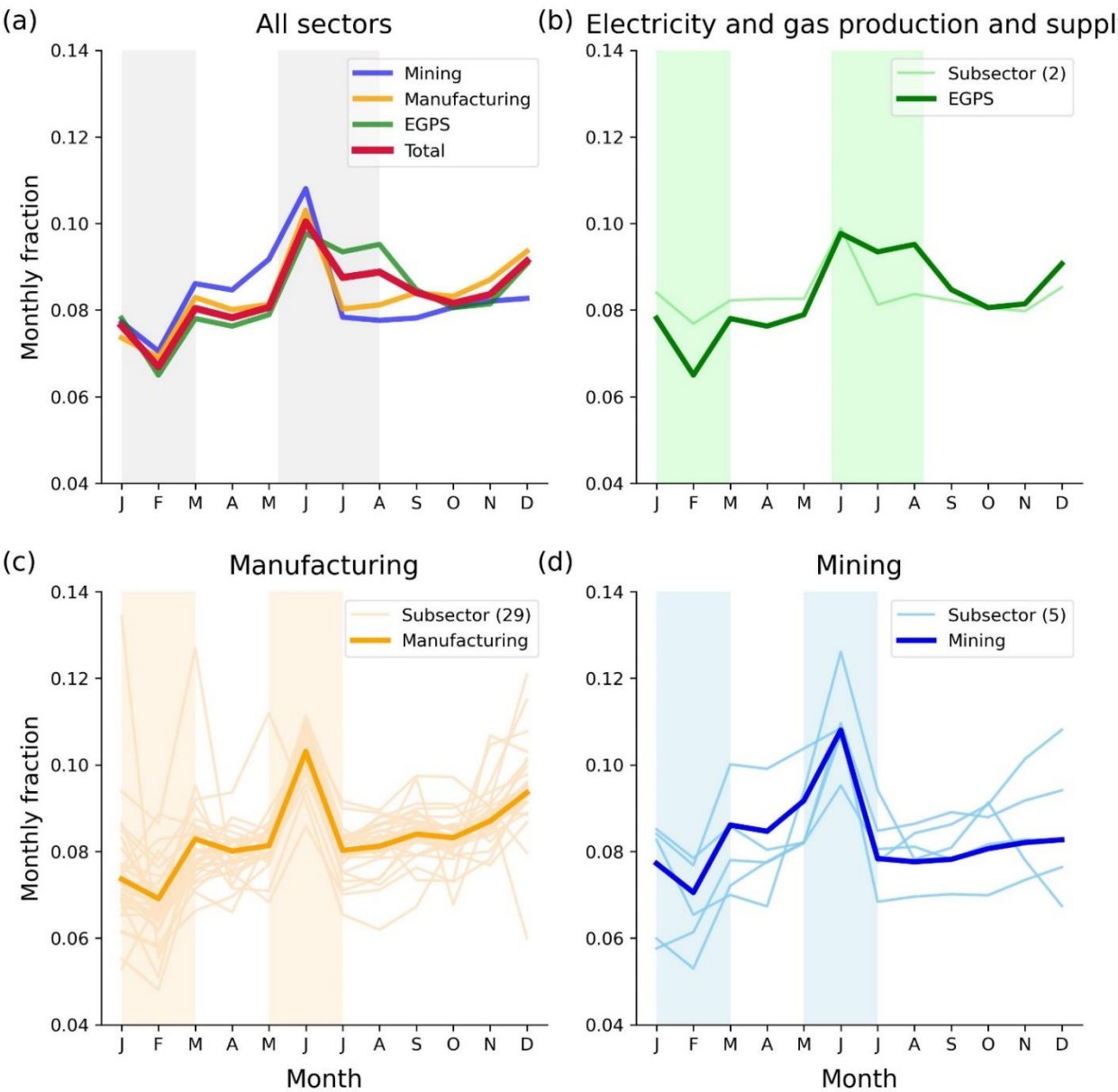

**Figure 4: Seasonal variations in the national total IWW (a) and for separate industrial sectors, including the electricity and gas production and supply (EGPS) (b), manufacturing (c), and mining sectors (d). The seasonal variations are represented as the fraction of the monthly IWW to the annual total during 2006-2010. The thick lines represent the water withdrawal of the main industrial sectors, and the thin lines represent the subsectors. The shadows represent the seasons with peak and low water withdrawal of a year.**

The seasonal variations in IWW during 2006-2010, represented by the fraction of monthly water withdrawal to annual total, are shown in Fig. 4. The results indicated that the IWW peaked in summer (June to August, 28%), followed by autumn (September to November, 25%), spring (March to May, 24%) and winter (December to February, 23%) (Fig. 4). February was the month with the lowest IWW, possibly due to its fewer days and the coincidence with the Chinese Spring Festival holiday (Liu et al., 2006). The highest IWW occurred in June, potentially due to the largest industrial output and high demand for cooling. This IWW peak did not extend to other summer months because extreme weather events, such as heatwaves and heavy rain, occurred more frequently in July and August, which could result in production shutdowns and reduced water consumption (Liu et al., 2006).

Seasonal patterns of IWW for the manufacturing and mining sectors were generally similar, but the subsectors of manufacturing showed more diverse patterns. The IWW for the EGPS had quite different seasonality, as there were two peaks, one in June to August and the other in December; these peaks were likely caused by the seasonal changes in cooling water withdrawal for thermal electricity generation due to seasonal temperature variation. The summer peak of EGPS was related to the high energy demand for air conditioning cooling (Huang et al., 2018), and the winter peak was related to the high energy demand for heating (Byers et al., 2014; Liu et al., 2015; Huang et al., 2018).

## 3.3 Long-term changes in industrial water withdrawal in China from 1965 to 2020

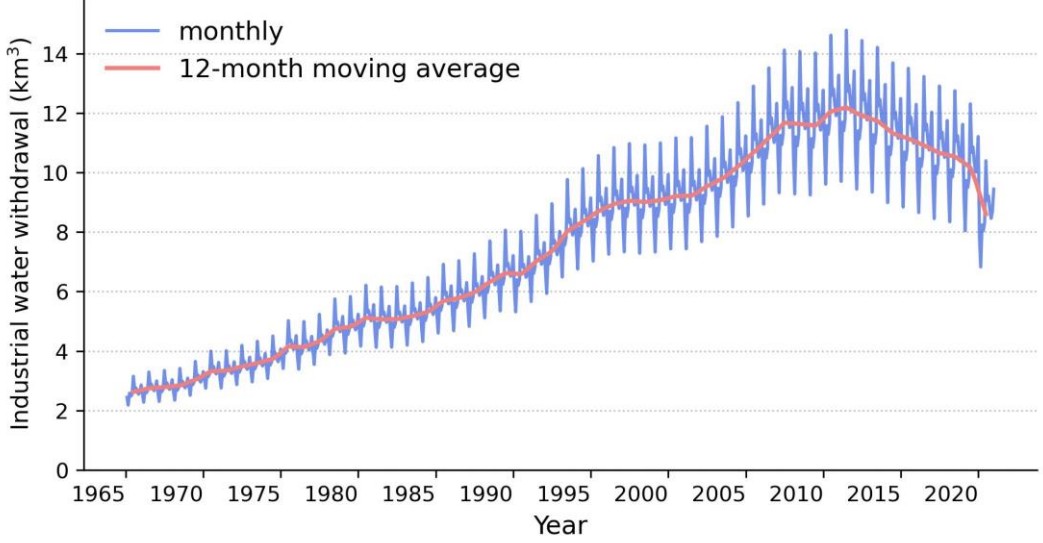

**Figure 5: Monthly industrial water withdrawal in China from 1965 to 2020 in the CIWW dataset. The red line represents the moving average of the monthly IWW of a 12-month moving window.**

For interannual monthly variations, IWW in China increased significantly from 2.1 billion to 14 billion m$^3$/month during 1965–2010, and it then decreased to 10 billion m$^3$/month (Fig. 5). These long-term changes indicated that IWW in China has now entered a slowly declining phase. The decline of national IWW after 2010 is mainly due to the implementation of a series of water-saving management measures (The State Council of the People's Republic of China, 2011) such as establishing "three red lines" to cap the total water withdrawal, enhance water use efficiency, and increase industrial water recycling rate(Chen and Chen, 2021; Zhang et al., 2023). In addition, the comparison of long-term annual national IWW of three datasets (CIWW, Huang and model data) showed that the other two datasets significantly underestimated China's total IWW and presented different temporal patterns, which could not consider the effects of water use policies (Fig. H8).

## 4 Discussion

Our study developed new gridded data for IWW in China from 1965 to 2020. The CIWW dataset improves upon previous data, particularly in the characterization of spatial and seasonal patterns. Instead of using indirect proxies, such as population density to map IWW, we used data on industrial enterprises that were direct water withdrawers. Compared with existing IWW data that either lack or only have limited representation of seasonal changes (Wada et al., 2011b; Huang et al., 2018; Brunner et al., 2019), our data contained the seasonal variations based on information from direct water consumers of the sectorial industrial production processes. Furthermore, we used localized data sources in China to produce the long-term IWW data, significantly improving regional accuracy and consistency with the statistical data records. The usage of public data sources and transparent methodology provide the possibility to further update and recalibrate the data for specific user needs.

**4.1 Potential applications of industrial water withdrawal data: high-resolution analysis and data scaling**

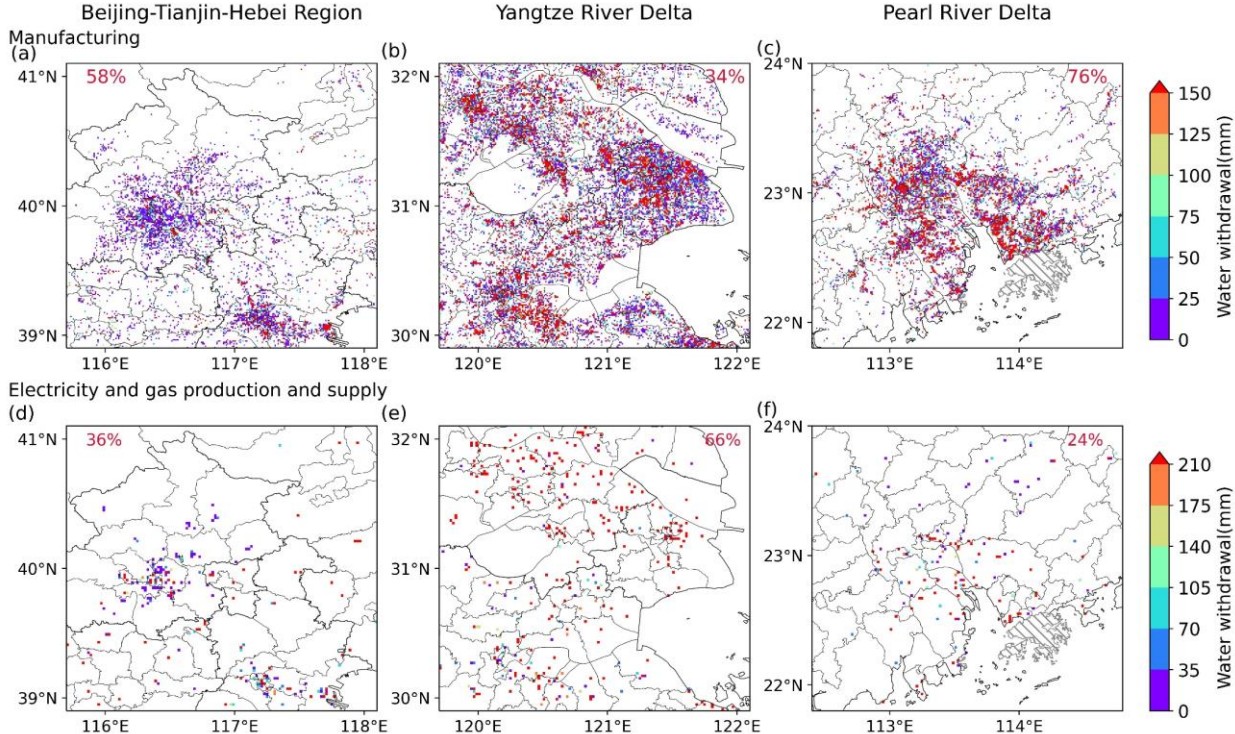

**Figure 6: Zoomed view of IWW in the densely urbanized regions in China at a spatial resolution of 0.01° (a, b, c) and 0.02° (d, e, f) for clarity, including the Beijing-Tianjin-Hebei region (a, d), Yangtze River Delta (b, e), and Pearl River Delta (c, f). Panels (a)–(c) show the spatial pattern of IWW for manufacturing, and Panels (d)–(f) show the spatial pattern of IWW for electricity and gas production and supply. The numbers displayed as percentages denote the percentage of the sectoral IWW to total IWW.**

The IWW data product with high resolution supports various research applications. The high spatial resolution showed IWW at fine scales. Figure 6 shows IWW hotspots in some of China's most densely urbanized regions in 2008 at 0.01° (this resolution was not included in the CIWW dataset but could be produced by the data and code we provided), including the Beijing-Tianjin-Hebei, the Yangtze River Delta, and the Pearl River Delta. These maps displayed high heterogeneity of IWW at the local scales.

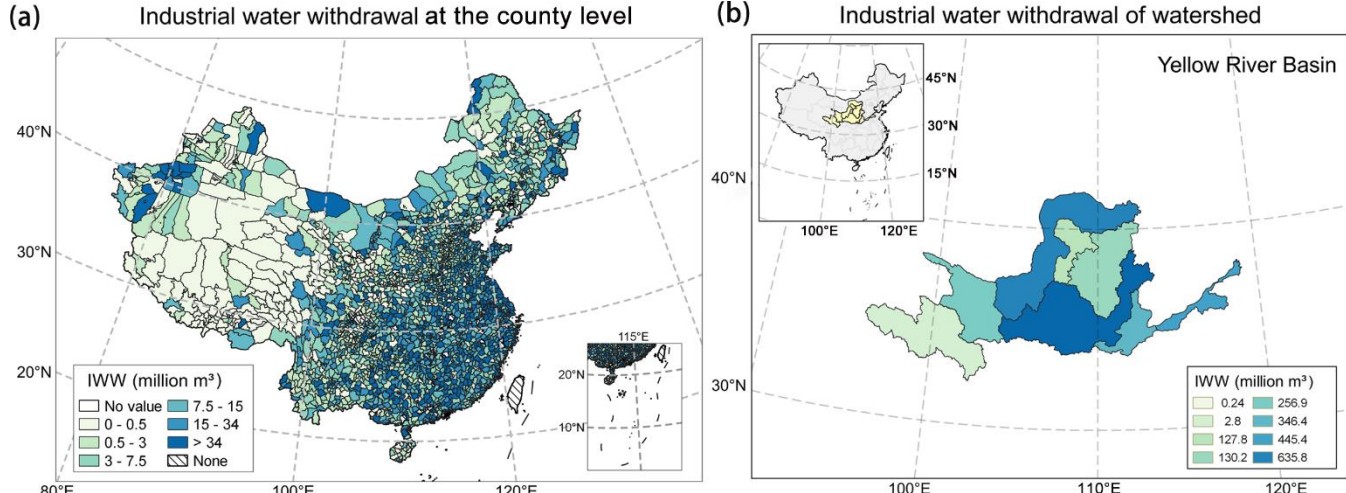

**Figure 7: CIWW data showing the downscaling of IWW from provincial to county levels in China (a) and from provincial to water basin levels in the Yellow River Basin (b).**

Additionally, CIWW data could facilitate downscaling of statistical data between different administrative (e.g., provincial or prefecture level), natural (e.g., watershed), and grid levels and help reconcile the scale mismatch between data with different spatial units (e.g., administrative and watershed/catchment). For example, with the gridded CIWW data, the statistical provincial IWW data could be downscaled to the prefecture level or even the county level (Fig. 7a). Moreover, the provincial IWW could be scaled to the watershed level using weights from the gridded IWW. Figure 7b shows the rescaling of the IWW from provincial levels to watersheds in the Yellow River basin.

## 4.2 Uncertainties in the spatial downscaling methods

The spatial pattern of IWW in the CIWW dataset was primarily derived based on >400,000 industrial enterprises in 2008. The spatial sampling of industrial enterprises could affect the spatial mapping. Although this was a large number of records, the enterprise dataset could not cover all enterprises in China since it only sampled enterprises above a designated production level. Therefore, other enterprises below this level, including their IWW, would be omitted from the datasets, leading to spatial undersampling of all industrial enterprises and their IWW in China. According to the 2008 Chinese Economic Census Yearbook, the enterprises above a designated level accounted for 93% of the IOV and 85% of the water withdrawal of all industries. This data indicated that spatial sampling could have a limited influence on the overall spatial pattern. Additionally, this issue could be mitigated when the point-level enterprise estimates were aggregated to the grid level.

Another source of uncertainty came from water use efficiency (WUE). Ideally, the enterprise-level IWW could be estimated using each enterprise's IOV and WUE. However, the enterprise-specific WUE was unavailable; thus, we used the provincial subsectorial WUE instead to estimate IWW, assuming the enterprises of the same subsector in the province had similar WUEs. This assumption disregarded the WUE variations since the WUE of different enterprises could vary substantially depending

on subsector, technological levels, investment, scale effects and so on. For this matter, the spatial distribution of IWW could be further improved with better data sources at finer scales in the future.

## 4.3 Uncertainties in seasonal allocation methods

When allocating the annual IWW to monthly scales, we used monthly variations in industrial product output data to represent the seasonal variation in IWW. Notably, there were differences in monthly variations across different products and provinces. When aggregating the monthly variations of 283 products to subsectors, each product was assigned an equal weight due to the lack of product-specific WUE, which neglected the structural differences within the subsector because the products consuming more water could have a more important role. When aggregating IWW from subsector to sector, the structural differences

within a sector were considered with the weights of subsector WUE.

We observed considerable differences in monthly variations in production output across provinces for different industrial sectors (Fig. I9). However, the seasonal fluctuations shown in sectors, such as manufacturing and mining, exhibited patterns that were chaotic and unreasonable at the provincial level (Fig. I9). It was difficult to determine whether these different seasonal fluctuations originated from statistical/random errors, unweighted product outputs to the subsector, interannual variability, or

actual regional differences. Therefore, we selected to use the national mean monthly variations to represent each subsector to improve the robustness. These monthly subsector variations were then combined with the subsectoral water withdrawal to derive the seasonal variations in IWW (Eq. 4). This choice was expected to have a limited impact on the seasonality of total IWW because it was primarily determined by the sector composition of a province (Reynaud, 2003; Sathre et al., 2022). In future research, the regional differences in seasonal variations in IWW should be further explored.

## 4.4 Uncertainties in producing long-term gridded data

A key step in developing the long-term gridded IWW data was to apply the spatial-seasonal pattern of IWW derived in 2008 for downscaling. The year 2008 was chosen to match the 2008 Chinese Economic Census Yearbook data, which include detailed IWW information that are only available in 2008. Thus, even though the total IWW increased over time with economic development, their spatial pattern and seasonality remained the same in CIWW. We acknowledge that the time-invariant

spatial-seasonal pattern of IWW from a single year in 2008 was a strong assumption and probably not true in reality. Nevertheless, this practice was acceptable in the literature under the data limit. For example, the spatial patterns from a single-year (e.g., the urban population distribution in 2009 used in WaterGAP3 and Global IWW map in 2000 used in PCR-GLOBAL) or patterns with multi-year updates (e.g., H08 and Huang et al., 2018) were used when developing the gridded IWW data with long time spans. Other time-varying data sources, such as nightlight, land cover, and population density maps with frequent

temporal updates, could potentially facilitate the characterization of the temporal changes in the spatial pattern of IWW.

The long-term changes in the industrial WUE can affect IWW, since WUE generally improves over time with the development of technology. This improvement would occur for all enterprises (Chen et al., 2019; Yang et al., 2021) and thus may not necessarily change the broad spatial pattern of IWW; this pattern is determined by the spatial distribution of industry and economic activities. The influence of other long-term factors such as climate change and WUE changes related to industry

development could be partially captured by the provincial statistical data which incorporate the changing spatial pattern of total IWW at the provincial level (Fig. I9).

Notably, the number of enterprises would also change over time and is likely to influence the spatial pattern of IWW. By comparing the spatial pattern of the IOV between 2008 and 2013 with the gridded enterprise data, the two years showed high consistency, with $R^2$ values of 0.9 at 0.1° and 0.94 at 0.25° (Fig. J10). Since the 2013 data had 16% fewer enterprise samples (<340,000) than 2008 (>400,000), the different sample sizes meant fewer enterprises would appear in 2013 compared to 2008. Nonetheless, the number of grids with the presence of valid enterprises in 2013 was just 12% fewer than that in 2008 at 0.1° and 7% at 0.25°, much smaller than the expected 16% decline in spatial coverage. This result indicated that the spatial pattern of the gridded data was less sensitive to the number of enterprises, especially at coarse spatial resolutions.

These analyses support the fact that specific industrial enterprises, their WUE, and water withdrawal substantially changed over time, and the broad spatial pattern after aggregating to the grid scale can still be applied because the spatial pattern of IWW is largely determined by the distribution of the population and economy of the country, which remain relatively stable over the years (Fig. G7). Nevertheless, temporal changes in the driving factors of IWW and their regional differences, such as industrial structure, water use efficiency, and climate (Alcamo et al., 2003; Otaki et al., 2008; Flörke et al., 2013; Zhou et al., 2020), should be considered to achieve higher accuracy. Due to this limitation, the CIWW dataset would have better performance for the last 20 years but may contain larger uncertainties towards earlier periods. Users can select the time period of the dataset according to their specific needs and interpret earlier years data with caution. Nevertheless, the CIWW data in earlier years showed surprisingly good performance with a much higher correlation (0.80 vs. 0.39~0.43 in 1971; as illustrated in Fig. K11) and smaller RRMSE (Relative Root Mean Squared Error, RMSE/mean, 1.81 vs. 2.67~ 2.76 in 1971) than other datasets when compared against Zhou2020 data at prefectural level (Note the prefecture-level IWW from Zhou2020 data was not used in developing CIWW).

## 5 Conclusions

To address the data gap in industrial water withdrawal in China, one of the top water consumers in the world, we developed a new gridded dataset, namely, the China Industrial Water Withdrawal Dataset. This dataset provided monthly IWW from 1965 to 2020 with spatial resolutions of 0.1° and 0.25°. With the best available data sources, this dataset showed significant improvements when compared to previous global datasets in characterizing the spatial pattern, seasonal variation, and long-term changes in IWW in China and had much higher accuracy. The transparent methodology and public availability of the source data enabled further adjustments and calibration to support the various applications by users. They also served as a reference to develop localized datasets for other countries. This dataset could help to understand human water use dynamics and support studies in hydrology, geography, environment, sustainability sciences, and regional water resource management and allocation in China.

## 6 Data availability

The China Industrial Water Withdrawal Dataset is available at https://doi.org/10.6084/m9.figshare.21901074 (Hou and Li, 2023). The Chinese Industrial Enterprises Database is available from the library resources of Peking University (https://www.lib.pku.edu.cn/portal/cn/news/0000001637). The Chinese Economic Census Yearbook in 2008 is freely available to the public at http://www.stats.gov.cn/tjsj/pcsj/jjpc/2jp/left.htm. The China Industry Product Output Database can be downloaded from the EPS data (https://www.epsnet.com.cn/). The provincial industrial water withdrawal data from 2003 to 2020 are from the China Water Resources Bulletin (http://www.mwr.gov.cn/sj/tjgb/szygb/), and the data from 1965 to 2002 were obtained from Zhou et al., 2020 (https://www.pnas.org/doi/10.1073/pnas.1909902117).

## Code availability

The Python codes used in this study are available at GitHub (https://github.com/cch-yhm/CIWW_dataset)

## Author contributions

Yan Li and Chengcheng Hou conceived and designed the study. Yinglu Liu and Chengcheng Hou contributed to data collection. Chengcheng Hou performed the data generation, data analysis and the original draft. Yan Li, Shan Sang, Xu Zhao, Yanxu Liu and Fang Zhao participated in reviewing and editing the paper. All authors have read and approved the paper.

## Competing interests

The contact author has declared that none of the authors has any competing interests.

## Financial support

This research is supported by the National Natural Science Foundation of China (no. 42041007, 41991235, and 72074136) and the Fundamental Research Funds for the Central Universities of China.

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

**Appendix A**

**Table A1 Classification of sectors in data**

| No. | Subsector | Sector | Notes |
|-----|-----------|--------|-------|
| 6 | Coal Mining and Dressing | Mining industry | |
| 7 | Petroleum and Naturel Gas Extraction | | |
| 8 | Ferrous Metals Mining and Dressing | | |
| 9 | Non-Ferrous Metals Mining and Dressing | | No industrial enterprise data |
| 10 | Nonmetal Minerals Mining and Dressing | | |
| 11 | Other Mining | | No monthly product output data, filled by average of mining sector |
| 13 | Food Processing | Manufacture industry | |
| 14 | Food Manufacture | | |
| 15 | Beverage Processing | | |
| 16 | Tobacco Processing | | |
| 17 | Textile Industry | | |
| 18 | Apparel, Footwear & Caps Manufacture | | |
| 19 | Leather, Furs, Down, and Related Products | | |
| 20 | Processing of Timber, Manufacturing of Wood, Bamboo, Rattan, Palm & Straw Products | | |
| 21 | Furniture Manufacturing | | |
| 22 | Paper & Paper Products | | |
| 23 | Printing, Reproduction of Recording Media | | |
| 24 | Cultural, Educational, and Sports Articles | | |
| 25 | Petroleum Processing and Coking | | |
| 26 | Raw Chemical Materials | | |
| 27 | Medicines Manufacturing | | |
| 28 | Chemical Fibres Manufacturing | | |
| 29 | Rubber Manufacturing | | |

| | | | |
|---|---|---|---|
| 30 | Plastics Manufacturing | | |
| 31 | Nonmetal Mineral Products | | |
| 32 | Smelting and Pressing of Ferrous Metal | | |
| 33 | Smelting and Pressing of Non-Ferrous Metal | | No industrial enterprise data |
| 34 | Metal Products | | |
| 35 | General Machinery | | |
| 36 | Special Machinery | | |
| 37 | Transportation Equipment | | |
| 39 | Electric Equipment and Machinery | | |
| 40 | Electronic and Telecommunications Equipment | | |
| 41 | Instruments, Metres, Cultural and Office Machinery | | |
| 42 | Artwork and Other Manufacturing Products | | |
| 43 | Waste Resources and Material Recycling and Processing | | No monthly product output data, filled by average of manufacturing sector |
| 44 | Electricity and Heating Power Production and Supply | Electricity and Gas Production and Supply | |
| 45 | Gas Production and Supply | | |
| 46 | Water Production and Supply | | Unused, not for industrial purpose Un used |


**Appendix B**

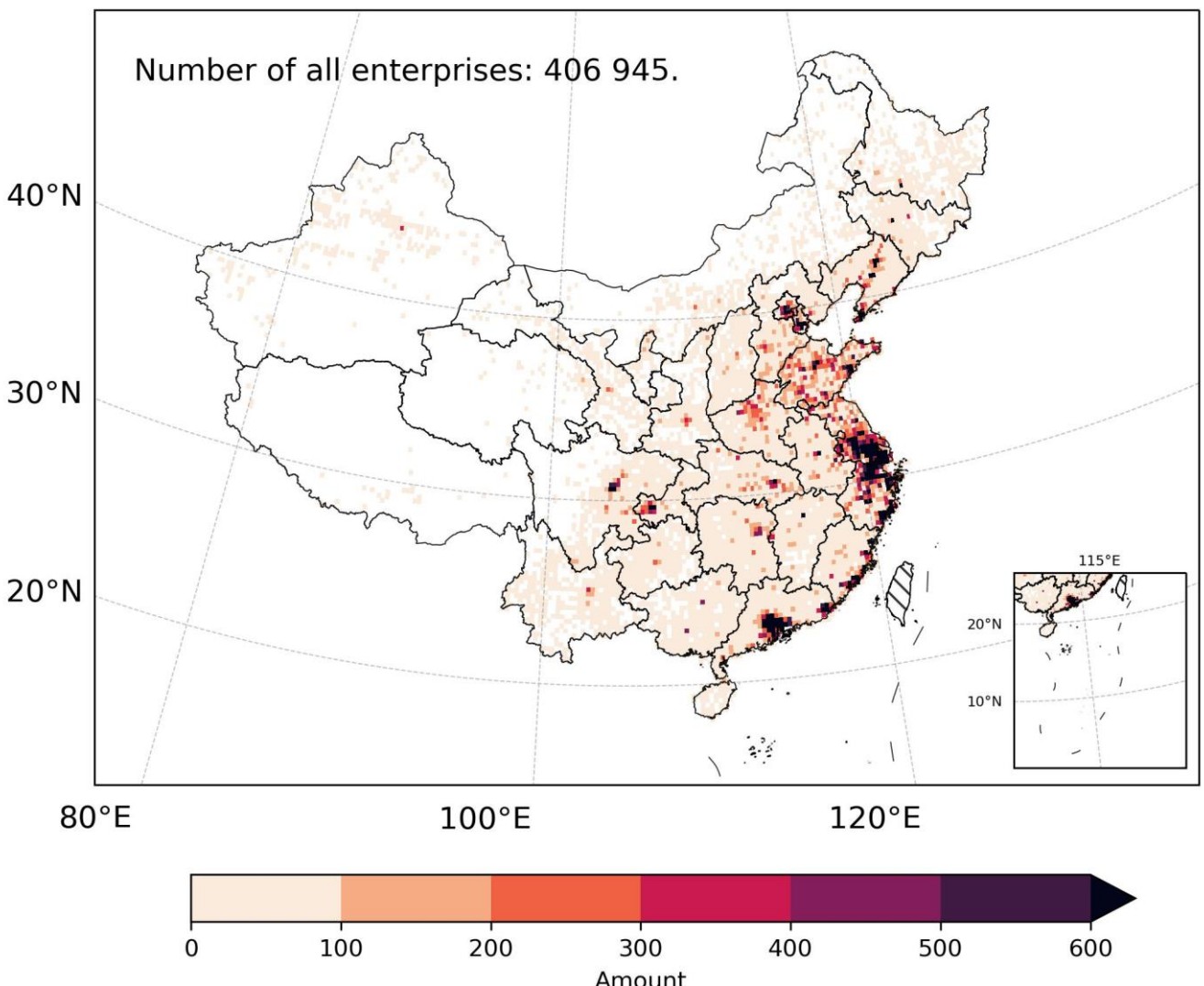

**Figure B2 Spatial distribution of the number of industrial enterprises in China at 0.25°.**

**Appendix C**

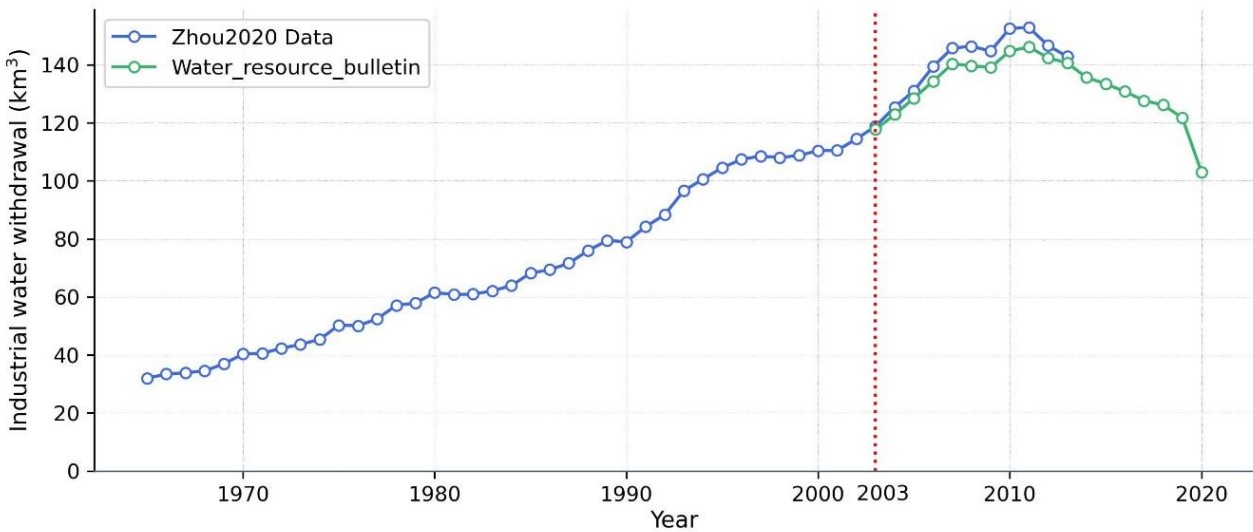


**Figure C3 The national IWW from two statistical data sources.**

**Appendix D**

**Table D4 A summary description of other IWW data for comparison.**

| Data variable | Data source | Industrial sector | Time span | Spatial resolution |
|---|---|---|---|---|
| Industrial water withdrawal | Global gridded monthly sectoral water use dataset | Sectors (3) | Monthly, 1971-2010 | 0.5˚ |
| Water abstraction for industrial uses | Input Data used in ISIMIP2b | None | Yearly, 1901-2005 | 0.5˚ |

**Introduction of IWW between different models in model data**

| IWW in model | Industrial sector | Definition of IWW |
|---|---|---|
| Water GAP | Sectors (2, except mining) | Total IWW is the sum of manufacturing and energy production water withdrawal |
| H08 | None | Total IWW includes manufacturing use and energy production. |
| PCR-GLOBWB | None | Total IWW no details available |


**Appendix E**

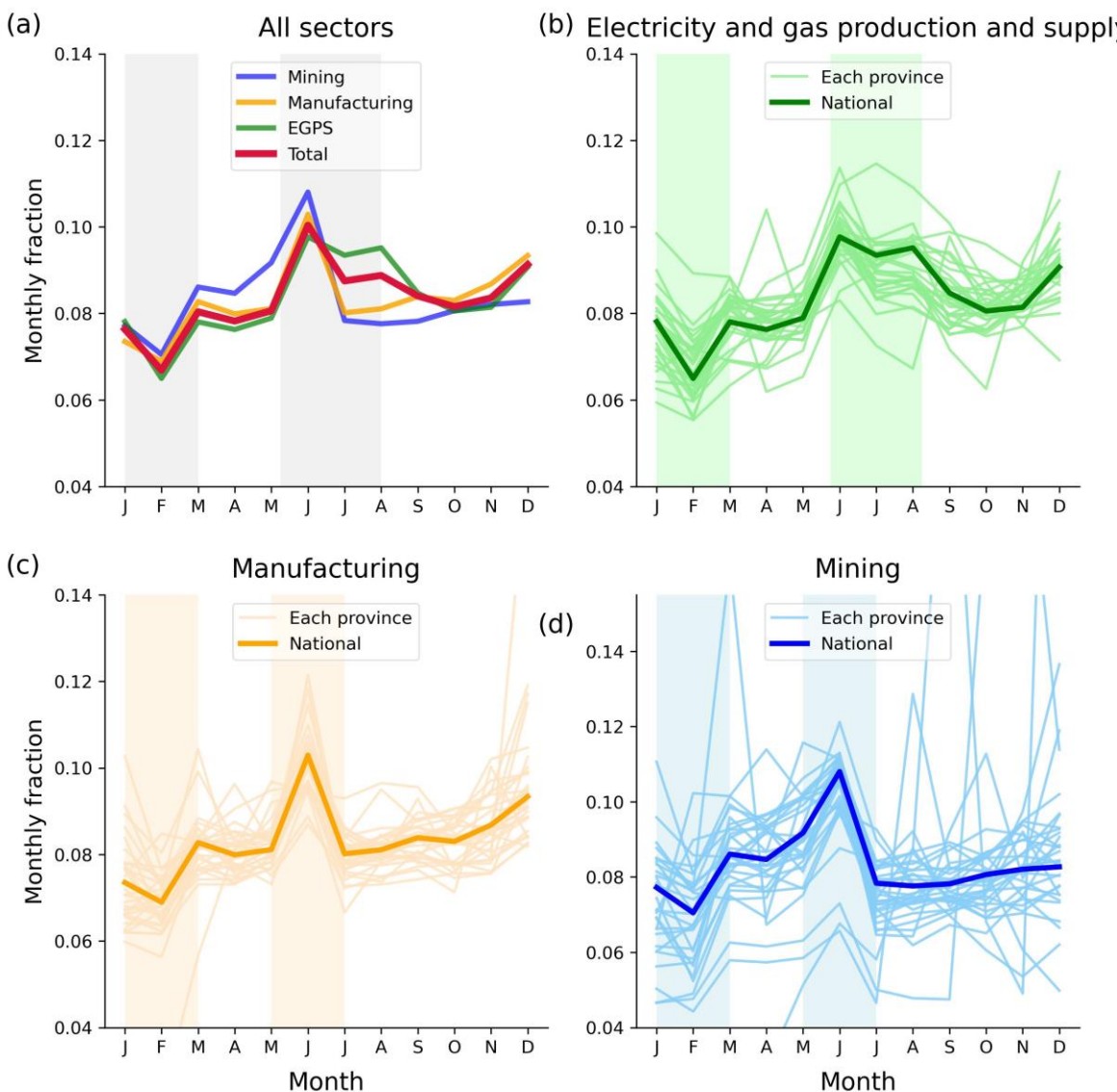

**Figure E5 Seasonal variations in the national total IWW (a) and provincial IWW for separate industrial sectors, including the electricity and gas production and supply (EGPS) (b), manufacturing (c), and mining sectors (d). The seasonal variations are represented as the fraction of monthly IWW to the annual total during 2006-2010. The thick lines represent the water withdrawal of sectors, and the thin lines represent provinces. The shadows represent the seasons with peak and low water withdrawals in a year.**


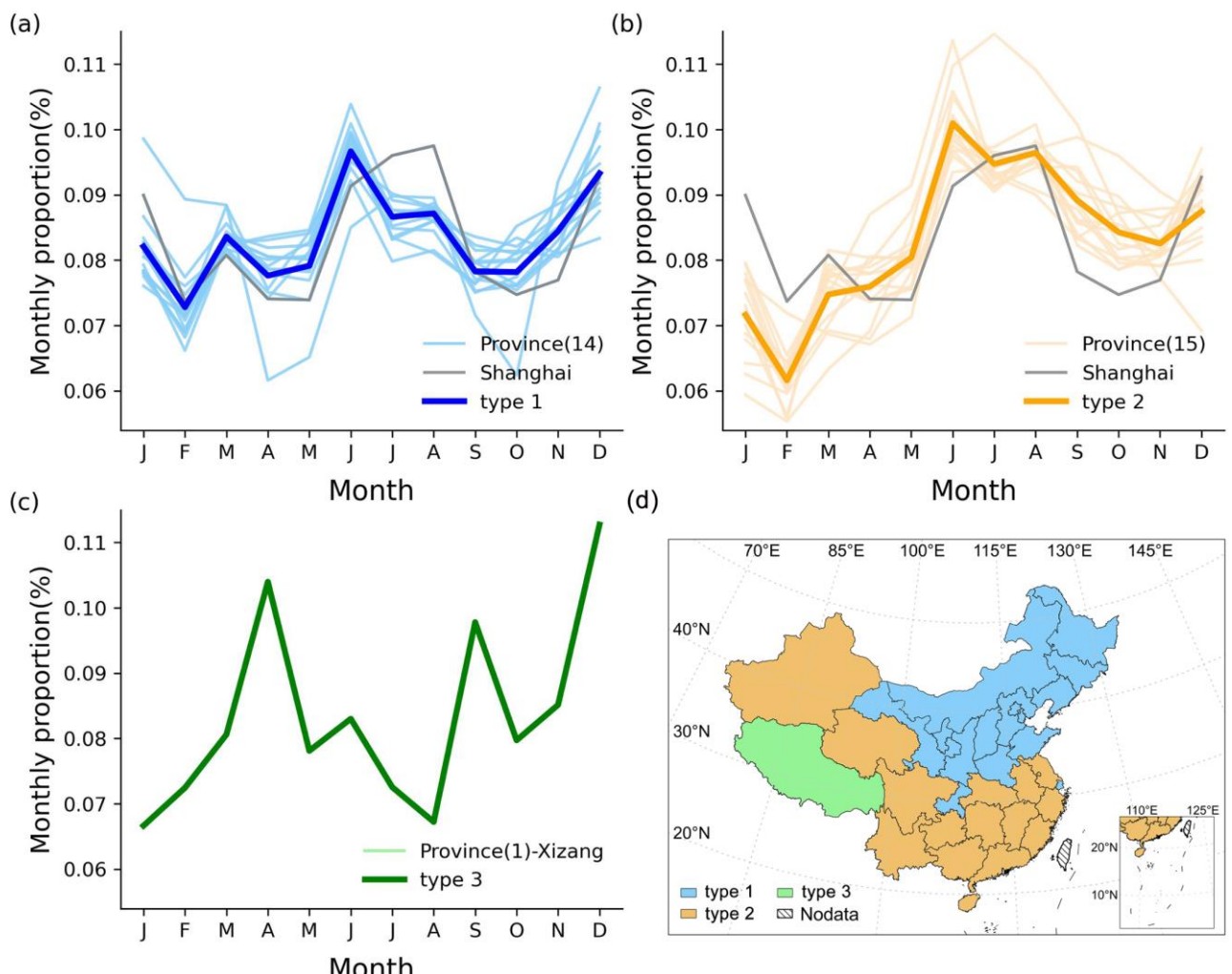

**Figure F6 Classification of seasonal variations of provincial water withdrawal of Electricity and Heating Power Production and Supply to type 1 (a), type 2 (b), type 3 (c), and d. map of three types. The seasonal variations are the faction of monthly IWW to the annual total during 2006-2010. The thick color lines show the mean seasonally of the cluster, and the thin color lines show the seasonally of each province.**


## Appendix G

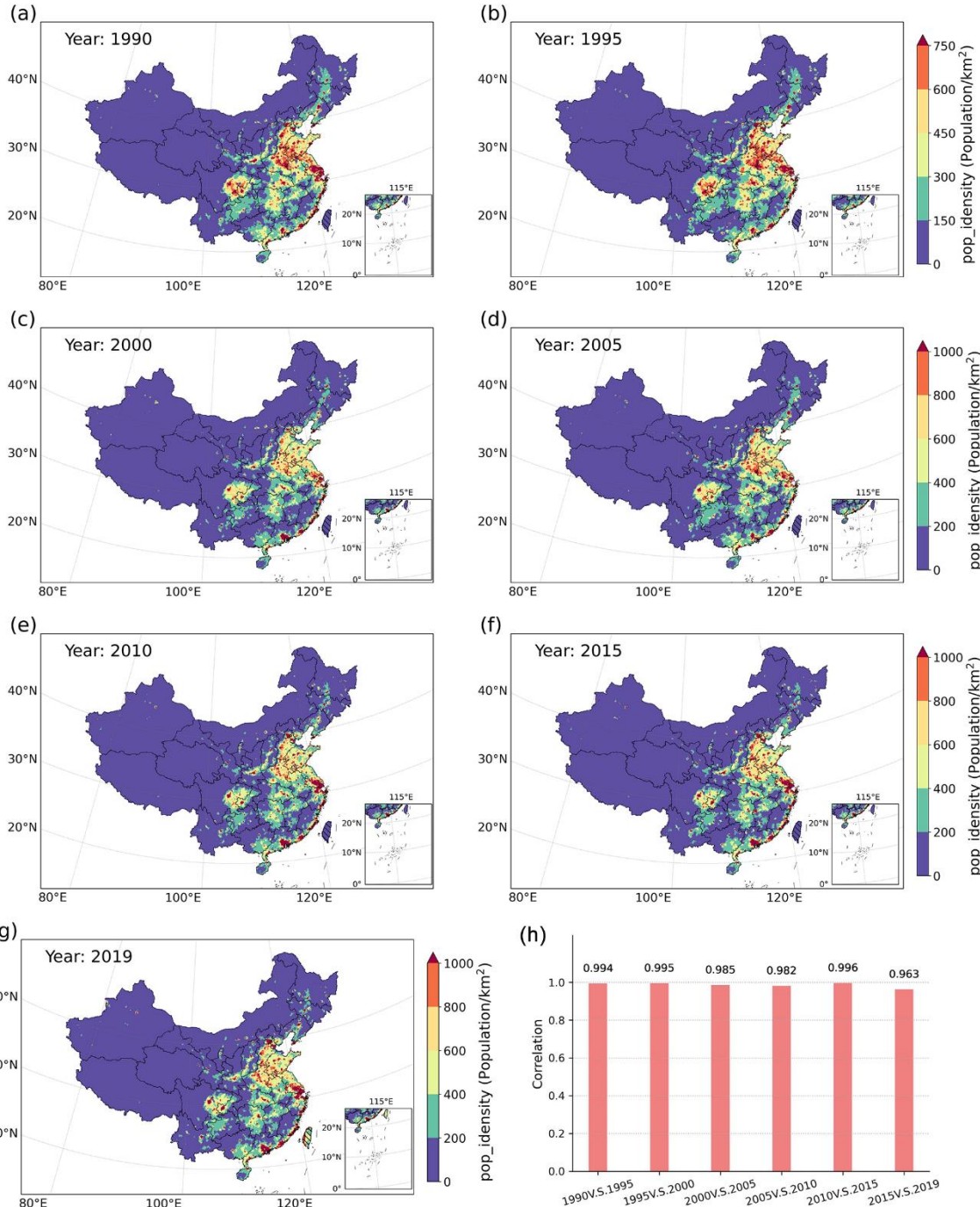

**Figure G7 The distribution pattern of population density in China from 1990 to 2019 (a-g) and Spearman's rank correlation coefficients of population density spatial pattern from 1990 to 2019 (h).**

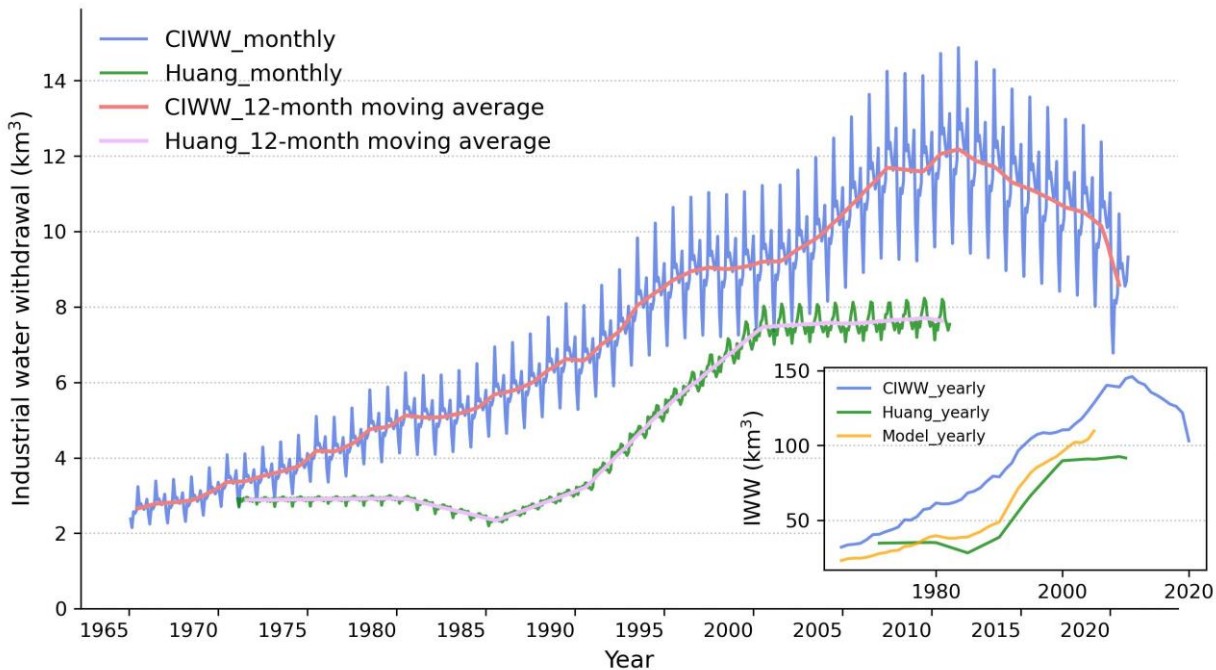

**Figure H8 Monthly and yearly IWW in China from 1965 to 2020 from the CIWW, Huang and model data.**

 **Appendix I**

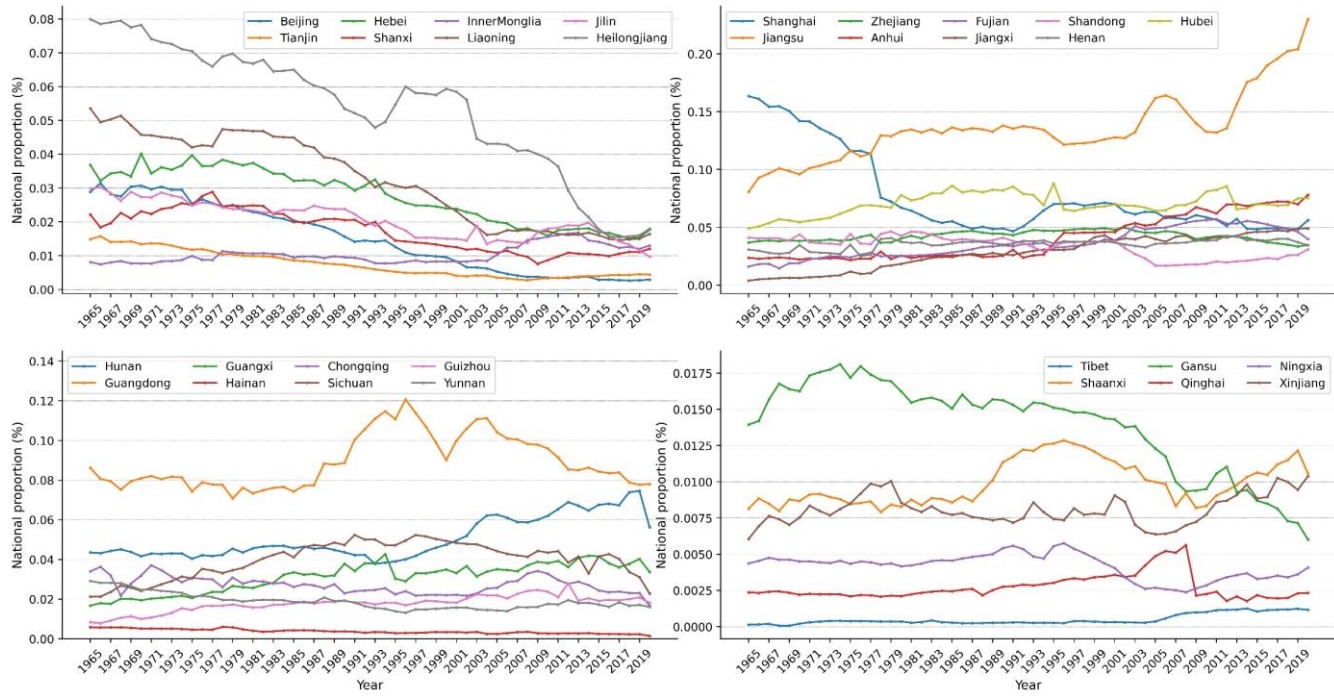

**Figure I9 The changing proportion of provincial IWW to national total from 1965 to 2020.**

**Appendix J**

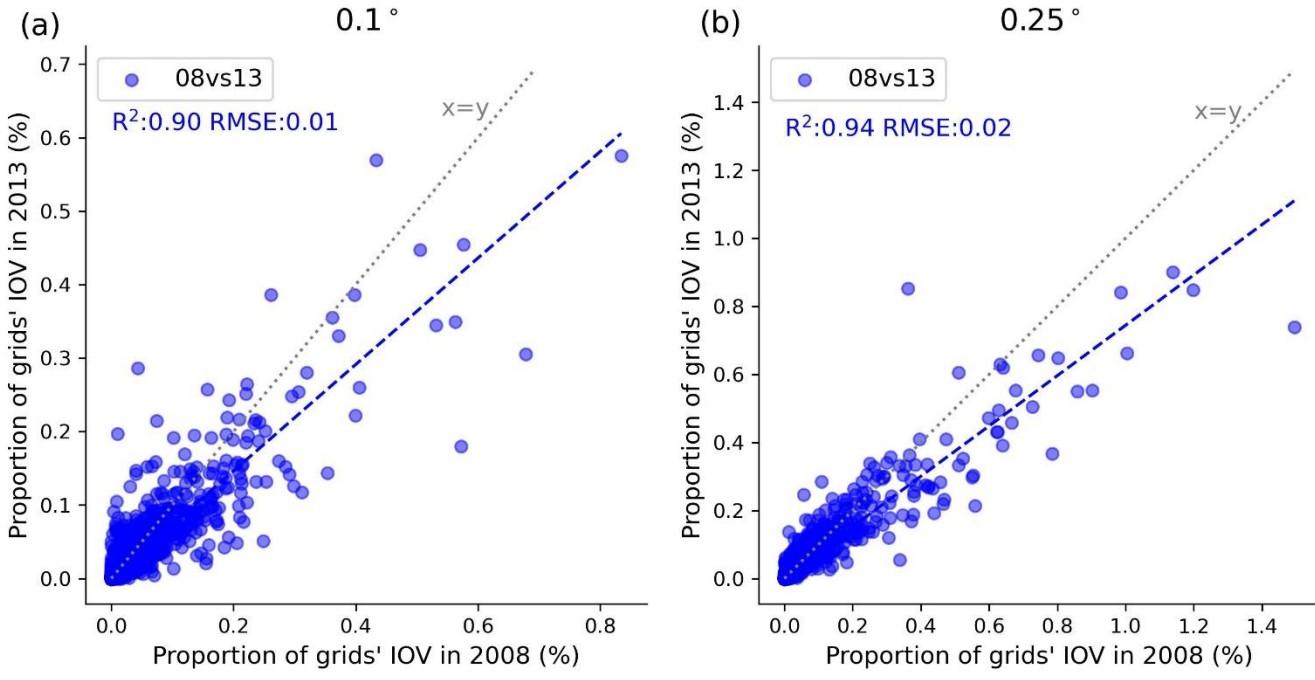


**Figure J10 Comparison of the spatial pattern of the IOV between 2008 and 2013 from the gridded enterprise data at (a) 0.1° and (b) 0.25°. The gridded IOV was normalized as the proportion of the gridded IOV of the country for comparison. The grey dotted line indicates the 1:1 line, and the blue dashed lines indicate the fitted lines.**

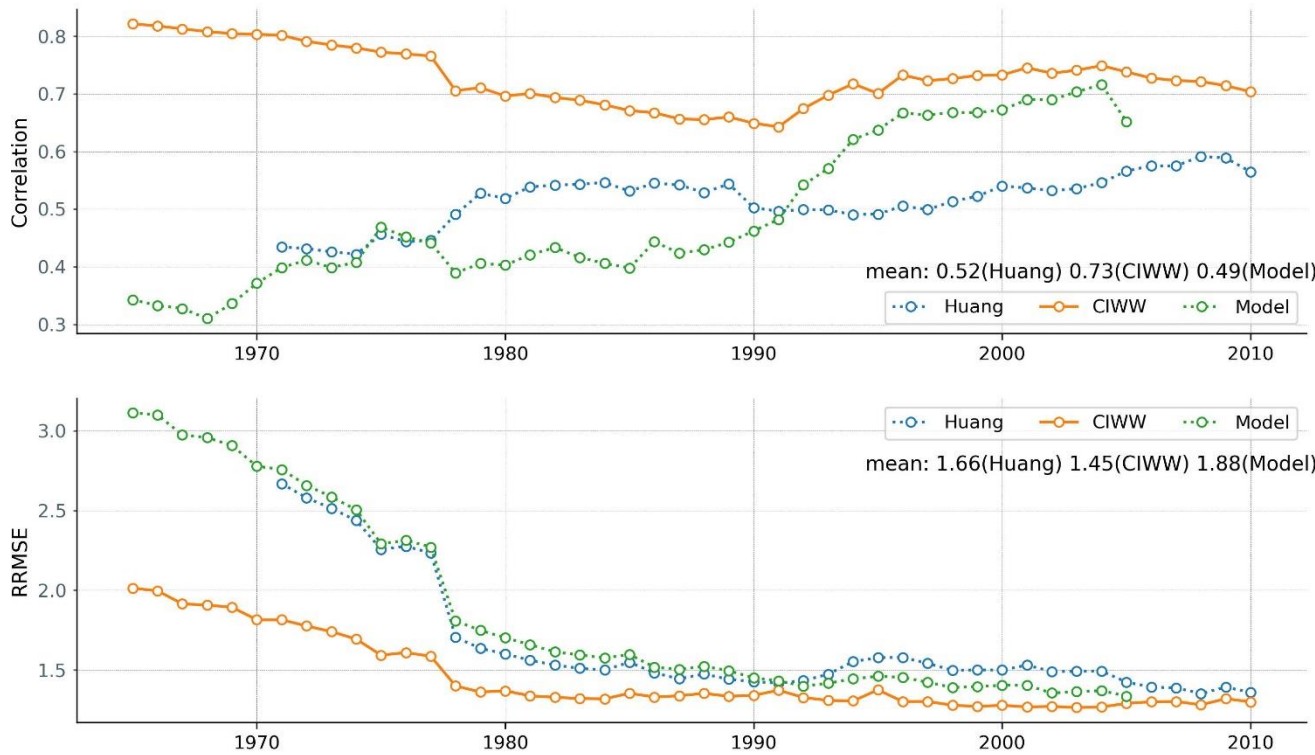

**Figure K11 Comparing the three gridded IWW (CIWW, Huang data, and model data) against Zhou2020 data at the prefecture-level from 1965 to 2010. Higher is better for correlation, and lower is better for RRMSE (Relative Root Mean Squared Error, RMSE/mean).**