# Peer review of "High-resolution mapping of monthly industrial water withdrawal in China from 1965 to 2020"

_Earth System Science Data, 2023_

## Author Comment (AC2)

**Response to Comments of Referee #2:**

**Hou et al. offer a new high-resolution gridded water industrial water withdrawal dataset for China, covering the period 1965 through 2020. The dataset relies on industry data not previously adopted in national water withdrawal datasets for China, and thus offer a promising contribution for water resources studies.**

**Response:** We would like to sincerely express our gratitude to you for your careful reading and constructive comments. According to the comments, we have tried our best to improve the manuscript, and an item-by-item response follows.

**The method is unclear, with some very questionable assumptions, including:**

1. **use of a constant water efficiency estimate from 2008 to extrapolate water withdrawal through time, not giving ample consideration to possible changes in industrial water use efficiencies and production per unit of water for each industry type since the 1960s.**

**Response:** Thank you for the comments.

We used constant water use efficiency and the resulting spatial-seasonal patterns of IWW in 2008 to downscale IWW in other years from 1965 to 2020, serving as a time-invariant pattern for downscaling. First, we made this choice mainly because of the data constraint since no data were available to calculate subsector water use efficiency for years other than 2008. This practice is not ideal but justifiable given the data limit and the practices adopted in other studies. Developing long-time series gridded data of IWW based on either a time-invariant pattern (e.g., H08, WaterGAP3, and PCR-GLOBAL) or patterns with decadal updates (e.g., Huang et al., 2018; Dong et al., 2022) for downscaling can be found in previous studies (Table R1). Second, industrial water use efficiency generally improved over time with the development of technology. This means that the temporal improvement in water use efficiency is likely to apply for all enterprises (Chen et al., 2019), while the spatial differences in water use efficiency of a given year are still determined by the spatial distribution of economic conditions which remain relatively stable over the years. The influence of long-term factors could be reflected through the changes in total IWW from statistical IWW data. For the above reasons, we chose the approach to develop the long-term gridded IWW data.

The dataset developed based on the time-invariant pattern of 2008 should be reasonably well for the recent ~20 years but may contain larger biases for earlier years. We added this important point to the manuscript to make the users aware of this issue so that they can choose the period of the data for their specific needs and accuracy considerations. In the revised manuscript, we added a more detailed discussion on using a time-invariant spatial pattern for long years IWW downscaling.

**Table R1. Spatial pattern used for long-term data extension in previous studies**

| Spatial pattern | Long-term data | Used for | Reference |
| --- | --- | --- | --- |
| NASA Back Marble night-time light | 1980-2016 | Model (VIC-5) | Droppers et al., 2020 |

| intensity map 2012-2016 | | | |
|---|---|---|---|
| Distribution of urban population in 2009 | 1950-2010 | Model (WaterGAP3) | Flörke et al., 2013 |
| Global population distribution map and national boundary information in 2005 | 1970-2010 | Model (H08) | Hanasaki et al., 2008 |
| Global IWW map in 2000 | 1960-2001 | Model (PCR-GLOBWB) | Wada et al., 2011a, b |
| Linear Interpolation based on GDP dataset in 1990, 2000 and 2010, same as 1990 before 1990 | 1971-2010 | Model (CLHMS, the Coupled Land Surface-Hydrologic Model System) | Dong et al., 2022 |
| Global population density maps with decadal updates (1980, 1990, 1995, 2000, 2005) | 1970-2010 | Water Dataset | Huang et al., 2018 |

**2. lack of units (e.g., unit of industrial output value?)**

**Response:** Thank you for the comments. We apologize for the confusion regarding the units of data variables. We have added the unit of industrial output value and revised the correctness of the units.

**3. unclear descriptions of existing water data used (the term "statistical data" is used throughout... does this mean "observed", "surveyed", "modeled" ...?);**

**Response:** Thank you for the comments. The statistical data for different variables could be collected through different methods.

We have supplemented the data types and sources as follows:

(a) 2.1.1 Surveyed data for industrial output value and water withdrawal

(b) The industrial enterprise dataset used in this study was from the Database of Chinese Industrial Enterprises in Mainland China from 1998 to 2013 (https://www.lib.pku.edu.cn/portal/cn/news/0000001637, last access: 18 May 2022), which is a surveyed dataset.

(c) 2.1.3 Surveyed data for monthly industrial product output

The monthly industrial product output data were from the China Industry Product Output Database (http://olap.epsnet.com.cn/auth/platform.html?sid=9C98BFB19A412FF66F744C2DA364ED5E_ipv473399501&cubeId=52, last access: 26 September 2021), which is also the surveyed data.

(d) 2.1.4 Statistical data and literature data for water withdrawal to extend long-term water withdrawal data

Provincial surveyed statistical data on IWW in China from the National Water Resources

Bulletin (http://www.mwr.gov.cn/sj/tjgb/szygb/, last access: 3 May 2022) from 2003 to 2020 were used.

**4. repeating seasonal pattern that fails to consider the important factors raised in the introduction, including weather and climate conditions changing though time.**

**Response:** Thank you for the comments. This is a good point.

The seasonal patterns derived from monthly industrial product output include signals of variations in climate and weather. For example, industrial product output for certain sectors could be affected by seasonal climate conditions and extreme weather events (e.g., production shutdowns or restrictions due to heat, thunderstorms, torrential rains, and other weather conditions). However, we acknowledge that climate change could affect the seasonality of industrial production and water consumption. For example, it is expected that a hotter summer increases cooling demands while a warmer winter reduces heating demands and thus affects IWW for energy production to some extent. However, we expect these seasonal changes would be slow and gradual, and their influence on IWW would be much smaller than the IWW increase driven by economic development. The seasonal and long-term climate change effects will be included in the yearly statistical IWW data. We added this supplement in the method section.

**5. Method must be described more clearly with all necessary details for a reader to reproduce the approach.**

**Response:** Thank you for the comments. We checked and revised the method section to clarify the key processes. Additionally, we will make the code and data open-access so that the readers can dig into more technical details and use them to reproduce our methods and results.

**6. The authors provide a limitations section that raises some of these problems. However, the assumptions lead to a dataset that fails to deliver on the stated goals of the manuscript--"a dataset with higher accuracy at fine scale." I would suggest either finding a way to extrapolate the more accurately, or reducing the ambition, so that the dataset not present water withdrawal from decades ago that is surely not credible given the assumptions used.**

**Response:** Thank you for the comments.

Our motivation for this work is that we realized that the currently available gridded datasets of IWW in China are those from global data which typically have poor performance at fine scales due to their methodology and data sources. Our data improve upon existing data in China mainly through two aspects: using spatial proxies that are directly relevant to IWW and localized statistical data sources for downscaling. Therefore, this study aims to develop a gridded long-term IWW dataset specifically for China with significant improvement upon other datasets at the local level to provide more accurate estimations of IWW. Following this suggestion, we revised the sentence in the abstract section:

"The CIWW dataset presented significant improvement upon previous studies in characterizing the spatial and seasonal patterns of IWW dynamics in China, ensuring consistency with statistical records at local scale."

To further validate the performance of the CIWW dataset, we added the spatial pattern comparison with the modeled data to the original comparison (Figure R1). Compared with the dataset from Huang data (Huang et al., 2018) and models data (ISIMIP2b Input Data-IWW-from 1901 to 2005, the average of three models (H08, PCR-GLOBWB, Water GAP)), our datasets is more consistent with prefecture literature data (Zhou et al., 2020), showing a much higher R2 (0.76, 0.47, and 0.56) and RMSE (Root Mean Square Error) (0.33 vs. 0.15 vs. 0.14 $10^9$m³) than Huang2018 data and model data. It means that the dataset improved industrial water withdrawal in China a lot.

Also, there are still uncertainties and limitations of the CIWW dataset. The dataset can be improved with better data sources in the future, especially early data. In the revision, we updated Figure R1 to Figure 2 and comparison results in section 2.3.

[Figure]

Figure R1 (Figure 2). Validation of CIWW data against statistical data for spatial distribution and seasonal variation. (a) The relationship between the mean IWW of 1971-2005 from literature data (Zhou et al., 2020) and CIWW, Huang2018 data (Huang et al., 2018) and model data (ISIMIP2b Input Data-IWW, average of three models (H08, PCR-GLOBWB, Water GAP) from 1901 to 2005) for 329 cities in China. The grey dotted line indicates the 1:1 line, and the colored dashed lines indicate the fitted lines. (b) The comparison of the 5-year mean (2006-2010) monthly variation in IWW from surveyed data (red, Long et al., 2020), CIWW (blue), and Huang2018 data (green) in Beijing. The solid grey line shows IWW for individual years from 2006 to 2010. The inset shows the annual mean total IWW from 2006 to 2010. For this comparison, CIWW was processed to the same spatial resolution of Huang2018 data at 0.5˚.

**Reference**

Chen, Q., Ai, H., Zhang, Y., and Hou, J.: Marketization and water resource utilization efficiency in China, Sustain. Comput. Inform. Syst., 22, 32–43, https://doi.org/10.1016/j.suscom.2019.01.018, 2019.

Dong, N., Wei, J., Yang, M., Yan, D., Yang, C., Gao, H., Arnault, J., Laux, P., Zhang, X., Liu, Y., Niu, J., Wang, H., Wang, H., Kunstmann, H., and Yu, Z.: Model Estimates of China's Terrestrial Water Storage Variation Due To Reservoir Operation, Water Resour. Res., 58, https://doi.org/10.1029/2021WR031787, 2022.

Droppers, B., Franssen, W. H. P., Van Vliet, M. T. H., Nijssen, B., and Ludwig, F.: Simulating human impacts on global water resources using VIC-5, Geosci. Model Dev., 13, 5029–5052, https://doi.org/10.5194/gmd-13-5029-2020, 2020.

Flörke, M., Kynast, E., Bärlund, I., Eisner, S., Wimmer, F., and Alcamo, J.: Domestic and industrial water uses of the past 60 years as a mirror of socio-economic development: A global simulation study, Glob. Environ. Change, 23, 144–156, https://doi.org/10.1016/j.gloenvcha.2012.10.018, 2013.

Hanasaki, N., Kanae, S., Oki, T., Masuda, K., Motoya, K., Shirakawa, N., Shen, Y., and Tanaka, K.: An integrated model for the assessment of global water resources - Part 2: Applications and assessments, Hydrol. Earth Syst. Sci., 12, 1027–1037, https://doi.org/10.5194/hess-12-1027-2008, 2008.

Huang, Z., Hejazi, M., Li, X., Tang, Q., Vernon, C., Leng, G., Liu, Y., Döll, P., Eisner, S., Gerten, D., Hanasaki, N., and Wada, Y.: Reconstruction of global gridded monthly sectoral water withdrawals for 1971-2010 and analysis of their spatiotemporal patterns, Hydrol. Earth Syst. Sci., 22, 2117–2133, https://doi.org/10.5194/hess-22-2117-2018, 2018.

Long, D., Yang, W., Scanlon, B. R., Zhao, J., Liu, D., Burek, P., Pan, Y., You, L., and Wada, Y.: South-to-North Water Diversion stabilizing Beijing's groundwater levels, Nat. Commun., 11, 3665, https://doi.org/10.1038/s41467-020-17428-6, 2020.

Wada, Y., Van Beek, L. P. H., Viviroli, D., Drr, H. H., Weingartner, R., and Bierkens, M. F. P.: Global monthly water stress: 2. Water demand and severity of water stress, Water Resour. Res., 47, 1–17, https://doi.org/10.1029/2010WR009792, 2011a.

Wada, Y., Beek, L. P. H. V., and Bierkens, M. F. P.: Modelling global water stress of the recent past : on the relative importance of trends in water demand and climate variability, Hydrol. Earth Syst. Sci., 15, 3785–3808, https://doi.org/10.5194/hess-15-3785-2011, 2011b.

Zhou, F., Bo, Y., Ciais, P., Dumas, P., Tang, Q., Wang, X., Liu, J., Zheng, C., Polcher, J., Yin, Z., Guimberteau, M., Peng, S., Ottle, C., Zhao, X., Zhao, J., Tan, Q., Chen, L., Shen, H., Yang, H., Piao, S., Wang, H., and Wada, Y.: Deceleration of China's human water use and its key drivers, Proc. Natl. Acad. Sci. U. S. A., 117, 7702–7711, https://doi.org/10.1073/pnas.1909902117, 2020.

---

## Author Response (AR1)

**Responses to Reviewers' Comments**

We deeply appreciate reviewers for the detailed and constructive comments. Following the helpful suggestions and comments, we have carefully revised the manuscript and provided a point-to-point response to each comment. The original comments are in **bold font**, our response is in regular font, and the changes in the text are in blue.

**Response to Comments of Referee #1:**

**This manuscript developed a gridded dataset of monthly industrial water withdrawal (IWW) for China, spanning a 56-year period from 1965 to 2020 at a spatial resolution of 0.1° and 0.25°. While the dataset covers a wide range of time, the spatial precision appears to be high. However, I have some concerns regarding the spatialization method used in the study. I think the method has too many uncertainties and strong assumptions, and the use of some definitions of industrial water is unclear. Therefore, I suggest that you make the following modifications to your manuscript to address these concerns:**

**Response**: Thank you for taking your time to review our study and provide feedback and comments. We appreciate your concerns regarding the spatialization method for industrial water withdrawal. Although far from perfect, we feel that using industrial enterprise data for spatialization has clear advantages compared to other spatial proxies such as population or nightlights, as the former is directly connected to industrial production processes in which water is withdrawn and consumed. In the revision, we performed more validations for the data and revised texts to clarify and discuss the assumptions and uncertainty in the methodology and data.

1. **The authors need to make the abstract more concise and focused. Instead of mentioning hydrology and geographical sustainability in a broad sense, the relevance of the dataset to specific research areas or applications should be emphasized.**

**Response:** Thank you for the comments.

We modified this part to be more specific, and the revised texts are shown below:

"The CIWW dataset, together with its methodology and auxiliary data, is useful for water resource management and hydrological models."

2. **Line35-45. The author lists the spatialization methods of sectoral IWW, but does not demonstrate the shortcomings of the current methods. The low accuracy of dataset is mentioned, but how the author judges the low accuracy of these datasets is not clear.**

**Response:** Thank you for the comments.

The shortcomings in current methods are the spatial proxies and the global statistical data used for downscaling, resulting in lower accuracy for regional applications. Firstly, the spatialization of IWW in manufacturing and mining relies on spatial proxies such as population density, urban or industrial area (e.g., Water GAP model 2.2 (Wada et al., 2016); Huang et al., 2018) (Table R1). These are only indirect factors related to IWW but not the factor that is directly relevant to industrial production processes that consume water (i.e., enterprise-level production). Moreover, they cannot separate different industrial sectors whose water use efficiency could be substantially different. Secondly, almost all existing IWW datasets are global datasets, which means they used national-level IWW statistical data downscaled to derive gridded data (Hejazi et al., 2014; Water GAP model2.2 (Wada et al., 2016); Huang et al., 2018)(Table R1), without incorporating information at sub-regional levels (e.g., provincial statistics). Therefore, global datasets are sufficient in revealing the global general pattern but may have poor performance for specific regions like China which keeps it from being used for localized studies (Liu et al., 2019). Besides, some IWW datasets are only estimated by water intake from electricity, omitting manufacturing and mining water withdrawal (e.g., H08 model (Wada et al., 2016)).

**Table R1 Method and data sources of IWW spatial mapping in previous studies**

| Sector | Method for Spatialization | Data sources | References |
|---|---|---|---|
| Total IWW | Downscaled only by demographic data | National data from World Resources Institute (country level) | WWDR-II Annual Industrial water withdrawal |
| | Downscaled by demographic data, socio-economic, and geographical data | FAO AQUASTAT database (country level) | Hanasaki et al., 2008a |
| | Downscaled by urban area data | | Otaki et al., 2008 |
| Total IWW only containing Electricity | Electricity production * Unit Water Demand Then downscaled by demographic data | Statistical data on Electricity production (country level) | H08 model (Wada et al., 2016) |
| | Electricity production * Unit Water Demand Then downscaled by demographic data, socio-economic, and geographical data | | Water GAP model 2.0 (Alcamo et al., 2007) |
| Electricity | Thermal electricity production * Unit Water Demand (point level) Then summed up to grid | Statistical data on Electricity production (Point level) | KASIM model (Koch and Vögele, 2009) Water GAP model 2.2 (Wada et al., 2016) Huang et al., 2018 |
| Manufacturing and mining | Downscaled only by demographic data | Statistical data (country level) | Water GAP model 2.2 (Wada et al., 2016) Huang et al., 2018 |
| | Total industrial water withdrawal - water withdrawal by electricity, omitting the mining water withdrawal | FAO AQUASTAT database (country level) | Hejazi et al., 2014 |

In the revision, we the revised texts and added more explanation for method introduction.

"Gridded datasets developed from administrative-level data or models provide more detailed spatial information (Hanasaki et al., 2008a; Wada et al., 2011a); however, their accuracy depends on the downscaling methods, including the spatial proxies and data sources. "

"For manufacturing, water withdrawal was estimated either as the residue of the energy water use from the total IWW downscaled using the spatial proxies mentioned above (Hejazi et al., 2014) or the product of population and per capita water consumption (Vörösmarty et al., 2000). Although several global gridded IWW datasets have been developed, the spatial proxies used for downscaling (e.g., population) are only indirect factors that are not directly tied to industrial production processes that consume water, and they cannot be used to separate the different industrial subsectors whose water use efficiency could be substantially different (0.32 of Paper and Paper Products versus 5.6 of Electric Equipment and Machinery, unit: 103 yuan/m3). Moreover, when downscaling, the global gridded datasets typically rely on the national statistical data (Hejazi et al., 2014; Water GAP model 2.2 (Wada et al., 2016); Huang et al., 2018) without incorporating subnational statistics to better capture the regional differences. Therefore, global datasets are sufficient in showing the global general pattern but can have poor performance for the specific regions, limiting their applications for regional water issues (Liu et al., 2019b)."

3. **line65-70. The rationale for the need for long-term and high-resolution IWW data in China requires further clarification. The reasons mentioned in the manuscript, such as water conflicts caused by increased water demand and water resource management are too broad and do not provide a specific explanation for the need of such data.**

**Response:** Thank you for the comments.

Industrial water withdrawal in China has been increasing, accounting for 20% of human water withdrawal, and shows substantial spatial variations. The gridded data of IWW are needed to characterize such changes for research and application purposes. However, the currently available gridded datasets of IWW in China are those from global data which typically have poor performance at fine scales due to their methodology and data sources. This is our motivation to specifically develop a gridded long-term IWW dataset in China with significant improvements in methodology and data sources compared to existing data to address the data gap.

With the gridded long-term IWW dataset in China, users can not only explore long-term changes of IWW, the tendency and effects of human water demand-supply in industrial activities at the local scale, and then provide recommendations on regional adjustment of industrial structure and water resources management; but also can be used as the reference and validation data applied to the model, with process-based models to gain an in-depth understanding of hydrological processes (Addor et al., 2020).

In the revision, we added a more detailed description on the reasons for the need.

"After decades of fast growth, China has become the second-largest economy in the world, with rapid industrial development leading to increasing water use (Zhou et al., 2020). IWW in China accounted for 20.2% of the total water withdrawal in 2019 (source: China Water Resources Bulletin) and increased by 4.5 times from 31.93 km3 in 1965 to 142.86 km3 in 2013 (Zhou et al., 2020). However, water resources in China are distributed unevenly in space, causing severe water stress due to a mismatch between the water supply and demand of the population and industrial development (Liu et al., 2013; Zhao et al., 2015). For instance, Northern China is one of China's largest industrial centres and densely populated regions, but it is experiencing the most severe water scarcity in the world (Yin et al., 2020). The changes in IWW and total water withdrawal have further increased the water conflict, making it urgent to optimize the current water use and management structure. Therefore, high-quality gridded IWW data for China are needed to characterize the spatial-temporal pattern of IWW for water management and for research on hydrological processes and modelling (Addor et al., 2020). However, IWW data produced from reliable data sources with a long period and high spatial resolution in China are still lacking. The publicly available data on IWW in China are either the statistical data at the provincial, prefecture, or basin level (Xia et al., 2017; Qin et al., 2020; Chen et al., 2021) or the gridded data extracted from the global datasets that have low accuracy for regional and local studies (Liu et al., 2019a, b; Han et al., 2019; Niva et al., 2020; Yin et al., 2020; Li et al., 2022)"

**4. Why should this sentence be placed here alone?**

**Response:** Thank you for pointing out this issue.

We want to emphasize that the data variable in this dataset is industrial water withdrawal rather than industrial water consumption. We have moved the sentence to the beginning of the introduction:

"Industrial water withdrawal (IWW) is the amount of water abstracted from freshwater sources for industrial purposes and does not consider water consumption; IWW accounts for approximately 19% of human water withdrawal globally and is the second largest sector of human water use following irrigation (WWAP, 2019)."

**5. In this manuscript, industrial water withdrawal and industrial water use are considered to have the same meaning. Nevertheless, the two definitions are different, and industrial water use also includes industrial reuse water consumption.**

**Response:** Thank you for the comments.

We apologize for the confusion regarding the definitions. The statistical IWW data from 2003 to 2020 were from the China National Water Resources Bulletin. The issue is that in China National Water Resources Bulletin (in Chinese), water withdrawal is called "water use" (in Chinese). However, according to its definition, it is defined as the annual amount of water withdrawal for industrial production activities, including primary production, auxiliary production, and ancillary production, excluding recycled water. Thus, the literal "water use" actually means "water withdrawal". We use the term "water withdrawal" when describing the

data sources to avoid this confusion.

To avoid confusion between the concepts of water withdrawal and water use, we replace water use with water withdrawal in Section 2.1.4 and add the definition as follows:

"IWW in the China Water Resources Bulletin is defined as the annual amount of water withdrawal for industrial production activities, including primary production, auxiliary production and ancillary production, excluding recycled water. To further extend the time series to an earlier period, the IWW reported by Zhou et al., (2020) (referred to as 'Zhou2020 data' hereafter) from 1965 to 2002, was used after summing the prefecture data to the provincial level; its IWW was defined the same way as the China Water Resource Bulletin and our study. Thus, the combination of the above two data sources provided complete statistical records of IWW from 1965 to 2020 in China."

6. **I think the spatialization method used has a lot of uncertainties. The authors assume the industrial water use efficiency was the same for all industrial enterprises in the same province and the same subsector. A province contains large, medium and small enterprises, and their water use coefficients must be different.**

**Response:** Thank you for the comments and we totally agree. In reality, the water use efficiency of a given enterprise could be different from other enterprises even for the same subsector, due to investment, technology, revenue, scale, and so on. It is reasonable to expect that enterprises of different sizes tend to have different water use efficiencies, and it is possible that larger companies may have higher water use efficiency than smaller ones. However, the problem is that currently we do not have data to provide specific information about the enterprise sizes and their water use efficiencies. If we arbitrarily introduce this scaling relationship without actual data, this would bring new uncertainty to spatial distribution. In the future, when such data becomes available, incorporating this information could better estimate enterprise-level IWW. We modified this part to be more specific in the revision:

"This assumption disregarded the WUE variations since the WUE of different enterprises could vary substantially depending on subsector, technological levels, investment, scale effects and so on. For this matter, the spatial distribution of IWW could be further improved with better data sources at finer scales in the future."

**Also, the distribution coefficient of monthly water shortage regards the whole country as a whole, without considering the differences among provinces.**

**Response:** Yes, there are indeed regional differences in the seasonality of IWW. Figure R1 shows the monthly variations of production output across provinces for different industrial sectors, and we can see most of them follow some differences in different provinces (e.g., for electricity and manufacturing sectors). However, at the provincial level, the seasonal fluctuations may exhibit unreasonable or chaotic patterns that are hard to explain, such as manufacturing and mining sectors of Hainan, Guangxi province. For example, Tibet's fraction of manufacturing production in January and February was too low, under 0.025. The exact

reasons are unclear, but they could be caused by statistical/random errors in the data. Therefore, we used each subsector's national mean monthly variations to allocate IWW instead of using provincial-level seasonal variations which are problematic in certain places. This choice would not affect much the seasonality of the final IWW because the seasonality of different sectors plays a dominant role in determining the seasonality of IWW for a province (Reynaud, 2003; Sathre et al., 2022).

[Figure]

**Figure R1 (Figure C3).** Seasonal variations in the national total IWW (a) and provincial IWW for separate industrial sectors, including the electricity and gas production and supply (EGPS) (b), manufacturing (c), and mining sectors (d). The seasonal variations are represented as the fraction of monthly IWW to the annual total during 2006-2010. The thick lines represent the water withdrawal of sectors, and the thin lines represent provinces. The shadows represent the seasons with peak and low water withdrawals in a year.

In the revised manuscript, we added Figure R1 as Figure 3C and a more detailed discussion on seasonal variations among provinces.

"We observed considerable differences in monthly variations in production output across provinces for different industrial sectors (Fig. C3). However, the seasonal fluctuations shown in sectors, such as manufacturing and mining, exhibited patterns that were chaotic and unreasonable at the provincial level (Fig. C3). It was difficult to determine whether these different seasonal fluctuations originated from statistical/random errors, unweighted product outputs to the subsector, interannual variability, or actual regional differences. Therefore, we selected to use the national mean monthly variations to represent each subsector to improve the robustness. These monthly subsector variations were then combined with the subsectoral water withdrawal to derive the seasonal variations in IWW (Eq. 4). This choice was expected to have a limited impact on the seasonality of total IWW because it was primarily determined by the sector composition of a province (Reynaud, 2003; Sathre et al., 2022). In future research, the regional differences in seasonal variations in IWW should be further explored."

**Moreover, the manuscript used the water use efficiency of enterprises in 2008 for the spatialization of IWW from 1965 to 2020. Can the coefficient of 2008 represent the period from 1965 to 2020?**

**Response:** We used water use efficiency and the resulting spatial-seasonal patterns of IWW in 2008 to downscale IWW in other years from 1965 to 2020, serving as a time-invariant pattern for downscaling. First, we made this choice mainly because of the data constraint since no data were available to calculate subsector water use efficiency for years other than 2008. This practice is not ideal but is justifiable given the data limit and the practices adopted in other studies. Developing long-time series gridded data of IWW based on either a time-invariant pattern (e.g., H08, WaterGAP3, and PCR-GLOBAL) or patterns with decadal updates (e.g., Huang et al., 2018; Dong et al., 2022) for downscaling can be found in previous studies (Table R2). Second, industrial water use efficiency generally improved over time with the development of technology. This means that the temporal improvement in water use efficiency is likely to apply for all enterprises (Chen et al., 2019), while the spatial differences in water use efficiency of a given year are still determined by the spatial distribution of economic conditions which remain relatively stable over the years. The changes in total IWW from statistical IWW data could reflect the influence of long-term factors. For the above reasons, we chose the approach to develop the long-term gridded IWW data.

The dataset developed based on the time-invariant pattern 2008 should be reasonably well for the recent ~20 years but may contain larger biases for earlier years. We added this vital point to the manuscript to make the users aware of this issue so that they can choose the period of the data for their specific needs and accuracy considerations. In the revised manuscript, we added a more detailed discussion on using a time-invariant spatial pattern for long years IWW downscaling.

**Table R2 Spatial pattern used for long-term data extension in previous studies**

| Spatial pattern | Long-term data | Used for | Reference |
| --- | --- | --- | --- |
| NASA Back Marble night-time light | 1980-2016 | Model (VIC-5) | Droppers et al., |

| | | | |
|---|---|---|---|
| intensity map in 2012-2016 | | | 2020 |
| Distribution of urban population in 2009 | 1950-2010 | Model (WaterGAP3) | Flörke et al., 2013 |
| Global population distribution map and national boundary information in 2005 | 1970-2010 | Model (H08) | Hanasaki et al., 2008 |
| Global IWW map in 2000 | 1960-2001 | Model (PCR-GLOBWB) | Wada et al., 2011a, b |
| Linear Interpolation based on GDP dataset in 1990, 2000 and 2010, same as 1990 before 1990 | 1971-2010 | Model (CLHMS, the Coupled Land Surface-Hydrologic Model System) | Dong et al., 2022 |
| Global population density maps with decadal updates (1980, 1990, 1995, 2000, 2005) | 1970-2010 | Water Dataset | Huang et al., 2018 |

We added a more detailed discussion on this matter in the revision:

Firstly, we modified this part to be more specific:

"Nevertheless, this practice was acceptable in the literature under the data limit. For example, time-invariant spatial patterns (e.g., H08, WaterGAP3, and PCR-GLOBAL) or patterns with decadal updates (e.g., Huang et al., 2018) were used when developing the gridded IWW data with long time spans. Other time-varying data sources, such as nightlight, land cover, and population density maps with frequent temporal updates, could potentially facilitate the characterization of the temporal changes in the spatial pattern of IWW. "

Besides, we added a discussion on the influence of the industrial WUE and stated the stable distribution of population and economy of the country:

"The long-term changes in the industrial WUE can affect IWW, since WUE generally improves over time with the development of technology. This improvement would occur for all enterprises (Chen et al., 2019; Yang et al., 2021) and thus may not necessarily change the broad spatial pattern of IWW; this pattern is determined by the spatial distribution of industry and economic activities. The influence of other long-term factors could be captured by the changes in the total IWW from the statistical data."

"These analyses support the fact that specific industrial enterprises, their WUE, and water withdrawal substantially changes over time, and the broad spatial pattern after aggregating to the grid scale can still be applied because the spatial pattern of IWW is largely determined by the distribution of the population and economy of the country, which remain relatively stable over the years."

Additionally, we specified limitations of the CIWW dataset for users:

"Due to this limitation, the CIWW dataset would have better performance for the last 20 years but may contain larger uncertainties towards earlier periods. Users can select the time period of the dataset according to their specific needs and interpret earlier years data with caution."

**Response to Comments of Referee #2:**

**Hou et al. offer a new high-resolution gridded water industrial water withdrawal dataset for China, covering the period 1965 through 2020. The dataset relies on industry data not previously adopted in national water withdrawal datasets for China, and thus offer a promising contribution for water resources studies.**

**Response:** We would like to sincerely express our gratitude to you for your careful reading and constructive comments. According to the comments, we have tried our best to improve the manuscript, and an item-by-item response follows.

**The method is unclear, with some very questionable assumptions, including:**

1. **use of a constant water efficiency estimate from 2008 to extrapolate water withdrawal through time, not giving ample consideration to possible changes in industrial water use efficiencies and production per unit of water for each industry type since the 1960s.**

**Response:** Thank you for the comments.

We used constant water use efficiency and the resulting spatial-seasonal patterns of IWW in 2008 to downscale IWW in other years from 1965 to 2020. We made this choice mainly because of the data constraint since no data were available to calculate subsector water use efficiency for years other than 2008.

Industrial water use efficiency generally improved over time with the development of technology. This means that the temporal improvement in water use efficiency is likely to apply for all enterprises (Chen et al., 2019), while the spatial differences in water use efficiency of a given year are still determined by the spatial distribution of economic conditions which remain relatively stable over the years. The influence of long-term factors could be reflected through the changes in total IWW from statistical IWW data. For the above reasons, we chose the approach to develop the long-term gridded IWW data.

The dataset developed based on the time-invariant pattern of 2008 should be reasonably well for the recent ~20 years but may contain larger biases for earlier years. We added this important point to the manuscript to make the users aware of this issue so that they can choose the period of the data for their specific needs and accuracy considerations. In the revised manuscript, we added a more detailed discussion on using a time-invariant spatial pattern for long years IWW downscaling.

We added a more detailed discussion on this matter in the revision:

 "The long-term changes in the industrial WUE can affect IWW, since WUE generally improves over time with the development of technology. This improvement would occur for all enterprises (Chen et al., 2019; Yang et al., 2021) and thus may not necessarily change the broad spatial pattern of IWW; this pattern is determined by the spatial distribution of industry and economic activities. The influence of other long-term factors could be captured by the changes

in the total IWW from the statistical data."

"These analyses support the fact that specific industrial enterprises, their WUE, and water withdrawal substantially changes over time, and the broad spatial pattern after aggregating to the grid scale can still be applied because the spatial pattern of IWW is largely determined by the distribution of the population and economy of the country, which remain relatively stable over the years."

Additionally, we specified limitations of the CIWW dataset for users:

"Due to this limitation, the CIWW dataset would have better performance for the last 20 years but may contain larger uncertainties towards earlier periods. Users can select the time period of the dataset according to their specific needs and interpret earlier years data with caution."

**2. lack of units (e.g., unit of industrial output value?)**

**Response:** Thank you for the comments. We apologize for the confusion regarding the units of data variables. We have added the unit of industrial output value and revised the correctness of the units.

**3. unclear descriptions of existing water data used (the term "statistical data" is used throughout... does this mean "observed", "surveyed", "modeled" ...?);**

**Response:** Thank you for the comments. The statistical data for different variables could be collected through different methods.

In the revision, we further modified the paper by only adding keywords to the sentences not amending the headlines in Section 2.1.

(a)2.1.1 Statistical data for industrial output value and water withdrawal

"The data included surveyed IOV and IWW for enterprises above a designated production level, consisting of three main industrial sectors (mining, manufacturing, and production and supply of electricity, gas and water) and 38 subsectors (Table A1)."

(b) Industrial enterprise data in China

"The dataset contains surveyed industrial information, including address, products, annual IOV, and industry category, for more than 400,000 enterprises whose annual IOV was more than 5 million Yuan (or 20 million Yuan from 2011 to 2013 due to standard changes)."

"To match the IWW survey data, which were only available in 2008, industrial enterprise data in 2008 were selected for spatial downscaling of the provincial IWW (Fig. B2)"

(d) 2.1.4 Statistical data of the industrial water withdrawal for long-term extension

"Provincial surveyed statistical data on IWW in China from the China Water Resources Bulletin (http://www.mwr.gov.cn/sj/tjgb/szygb/, last access: 3 May 2022) from 2003 to 2020 were used."

"To further extend the time series to an earlier period, the IWW reported by Zhou et al., (2020) (referred to as 'Zhou2020 data' hereafter) from 1965 to 2002, was used after summing the prefecture data to the provincial level."

Besides, we modified the descriptions of data in Section 2.3 of the paper:

"model data (ISIMIP2b Input Data-IWW, average of three models (H08, PCR-GLOBWB, Water GAP) from 1901 to 2005"

"monthly surveyed statistical IWW data in Beijing from 2006 to 2010 (Long et al., 2020)"

"However, the magnitude of IWW was significantly overestimated by the Huang2018 data (56 mm per year) relative to the surveyed statistical data (33 mm per year). In comparison, the magnitude of IWW in the CIWW data (31 mm per year) was more in line with the surveyed statistical data (Fig. 2b)."

4. **repeating seasonal pattern that fails to consider the important factors raised in the introduction, including weather and climate conditions changing though time.**

**Response:** Thank you for the comments. This is a good point.

The seasonal patterns derived from monthly industrial product output include signals of variations in climate and weather. For example, industrial product output for certain sectors could be affected by seasonal climate conditions and extreme weather events (e.g., production shutdowns or restrictions due to heat, thunderstorms, torrential rains, and other weather conditions). However, we acknowledge that climate change could affect the seasonality of industrial production and water consumption. For example, it is expected that a hotter summer increases cooling demands while a warmer winter reduces heating demands and thus affects IWW for energy production to some extent. However, we expect these seasonal changes would be slow and gradual, and their influence on IWW would be much smaller than the IWW increase driven by economic development. The seasonal and long-term climate change effects will be included in the yearly statistical IWW data.

We added this supplement in the method section.

"The seasonal pattern included signals of variations in climate and weather because the industrial product output for some sectors could be affected by seasonal climate conditions and extreme weather events (e.g., production shutdowns or restrictions due to heatwaves, thunderstorms, torrential rains). Since the climate change-induced seasonality changes were slow and gradual, their influences on monthly IWW were also low, and the long-term climate change impacts (e.g., warming) could be captured by the yearly statistical IWW data."

5. **Method must be described more clearly with all necessary details for a reader to reproduce the approach.**

**Response:** Thank you for the comments. We checked and revised the method section to clarify the key processes. Additionally, we will make the code and data open-access so that the readers can dig into more technical details and use them to reproduce our methods and results.

6. **The authors provide a limitations section that raises some of these problems. However, the assumptions lead to a dataset that fails to deliver on the stated goals of**

**the manuscript--"a dataset with higher accuracy at fine scale." I would suggest either finding a way to extrapolate the more accurately, or reducing the ambition, so that the dataset not present water withdrawal from decades ago that is surely not credible given the assumptions used.**

**Response:** Thank you for the comments.

Our motivation for this work is that we realized that the currently available gridded datasets of IWW in China are those from global data which typically have poor performance at fine scales due to their methodology and data sources. Our data improve upon existing data in China mainly through two aspects: using spatial proxies that are directly relevant to IWW and localized statistical data sources for downscaling. Therefore, this study aims to develop a gridded long-term IWW dataset specifically for China with significant improvement upon other datasets at the local level to provide more accurate estimations of IWW. Following this suggestion, we revised the sentence in the abstract section:

"Our CIWW dataset was significantly improved in comparison to previous data for the characterization of the spatial and seasonal patterns of the IWW dynamics in China and showed consistency with statistical records at the local scale."

To further validate the performance of the CIWW dataset, we added the spatial pattern comparison with the modeled data to the original comparison (Figure R1). Compared with the dataset from Huang data (Huang et al., 2018) and models data (ISIMIP2b Input Data-IWW-from 1901 to 2005, the average of three models (H08, PCR-GLOBWB, Water GAP)), our datasets is more consistent with prefecture literature data (Zhou et al., 2020), showing a much higher R2 (0.71, 0.54, and 0.62) and lower RMSE (Root Mean Square Error) (0.32 vs. 0.37 vs. 0.36 km³) than Huang2018 data and model data. It means that the dataset improved industrial water withdrawal in China a lot.

Also, there are still uncertainties and limitations of the CIWW dataset. The dataset can be improved with better data sources in the future, especially early data.

[Figure]

**Figure R1 (Figure 2).** Validation of the CIWW data against the statistical data for spatial

distribution and seasonal variation. (a) Relationship between the mean IWW of 1971-2005 from Zhou2020 data (Zhou et al., 2020) and CIWW, Huang2018 data (Huang et al., 2018) and model data (ISIMIP2b Input Data-IWW, average of three models (H08, PCR-GLOBWB, Water GAP) from 1901 to 2005) for 329 cities in China. The black dotted line indicates the 1:1 line, and the colored dashed lines indicate the fitted lines. For this comparison, CIWW is processed to the same spatial resolution of Huang2018 data and model data at 0.5° before aggregating to the prefecture level. Comparison results with CIWW at other resolutions (0.25° and 0.1°) are reported in $R^2$ and RMSE. (b) Comparison of the 5-year mean (2006-2010) monthly variation in IWW from the surveyed data (red, Long et al., 2020), CIWW (blue), and Huang2018 data (green) in Beijing. The solid grey line shows IWW for individual years from 2006 to 2010. The inset shows the annual mean total IWW from 2006 to 2010.

In the revision, we updated Figure R1 to Figure 2 and comparison results in section 2.3.

"For spatial validation, the 35-year mean IWW (1971-2005) from CIWW, global gridded data (Huang et al., 2018) (referred to as Huang2018 data), and model data (ISIMIP2b Input Data-IWW, average of three models (H08, PCR-GLOBWB, Water GAP) from 1901 to 2005 with units converted from $m^3$ to mm) were compared with the Zhou2020 data (treated as "truth") (Zhou et al., 2020) for 329 prefectures in China. Although we used the Zhou2020 data at the provincial level to produce the CIWW dataset, the validation here was at the prefecture level, whose information was unused by CIWW. The validation at the prefecture level can determine how the effectiveness of spatial patterns after downscaling. All gridded data were averaged over each prefecture using the *rasterstats* package in Python and then multiplied by the prefecture area to obtain IWW for each prefecture (in units of km³). The results in Fig. 2a indicated a superior performance of CIWW data in representing the spatial variations in IWW compared against Huang2018 data and model data due to its much higher $R^2$ values (0.71, 0.54, and 0.62) and lower root mean square error (RMSE) (0.32 vs. 0.37 vs. 0.36 km³). Additionally, when comparing CIWW at higher resolutions (0.25° and 0.1°), the consistency with the Zhou2020 data improved further with higher $R^2$ values (0.79 and 0.78, respectively) than the 0.5° data. This result demonstrated the benefit of increased spatial resolution in characterizing the IWW at smaller scales."

**Response to Comments of Referee #3:**

**This paper aims to create a monthly-scale spatial distribution dataset of China's industrial water use over the past 40 years. While the method of spatial downscaling in the article is clear, there are several key issues that require further explanation.**

**Response**: We would like to sincerely express our gratitude for the careful reading and constructive comments, and we are trying our best to address these raised issues and make revisions.

1. **Although the author discusses the uncertainty, it is not enough to be convinced. For instance, the author proposed that "We found that the spatial pattern from the 1998 data was similar to 2008 at 0.25°". However, the Pearson correlation can only reflect the correlation between the variables. The author must use other statistical indicators to compare the differences between two years. In addition, the years span from 1965 to 2020. Would the enterprise data for 35 years be not changed?**

**Response:** Thank you for the helpful suggestion!

After aggregating a large number of enterprises to grid levels, the spatial pattern of IWW from the gridded data reflected the broad-scale industry distribution and was not sensitive to specific enterprises. The number of enterprises will change over time for sure. However, the reflected spatial pattern of industry distribution is relatively stable as it is also linked to the overall population and economy of the country.

To demonstrate this point, we estimated enterprises' spatial coverage and industrial output value (IOV) between 2008 and 2013. We did not use data in 1998 as the 1998 data contained significantly fewer enterprises (<170,000) compared to 2008 (>400,000), while the 2013 data included about 340,000 enterprises. The different sample sizes between the two years would undoubtedly affect the comparison of their spatial patterns, meaning fewer enterprises would appear in 2013 compared to 2008 (~16% fewer samples than in 2008). Nonetheless, the number of grids in 2013 with valid enterprise is just 12% fewer than in 2008 at 0.1° and 7% at 0.25°. This suggests that the spatial pattern of gridded data is not sensitive to the number of enterprises, especially at coarse spatial resolutions. Otherwise, a 16% reduction in the sample would lead to a similar magnitude of reduction in the valid grids. Moreover, we compared gridded IOV between two years. Results indicated a high consistency between 2008 and 2013 with R2 of 0.90 at 0.1° and 0.94 at 0.25° (Figure R1). These comparisons further support the fact that although specific enterprises would change over time, the spatial pattern of industry and IWW reflected by hundreds of thousands of enterprises remained stable.

[Figure]

**Figure R1 (Figure D4).** Comparison of the spatial pattern of the IOV between 2008 and 2013 from the gridded enterprise data at (a) 0.1° and (b) 0.25°. The gridded IOV was normalized as the proportion of the gridded IOV of the country for comparison. The grey dotted line indicates the 1:1 line, and the blue dashed lines indicate the fitted lines.

In the revision, we added Figure R1 as Figure D4 in the manuscript to further illustrate the consistency of the spatial pattern of industry and IWW. We also added a more detailed discussion.

"Notably, the number of enterprises would also change over time and is likely to influence the spatial pattern of IWW. By comparing the spatial pattern of the IOV between 2008 and 2013 with the gridded enterprise data, the two years showed high consistency, with $R^2$ values of 0.9 at 0.1° and 0.94 at 0.25° (Figure D4). Since the 2013 data had 16% fewer enterprise samples (<340,000) than 2008 (>400,000), the different sample sizes meant fewer enterprises would appear in 2013 compared to 2008. Nonetheless, the number of grids with the presence of valid enterprises in 2013 was just 12% fewer than that in 2008 at 0.1° and 7% at 0.25°, much smaller than the expected 16% decline in spatial coverage. This result indicated that the spatial pattern of the gridded data was less sensitive to the number of enterprises, especially at coarse spatial resolutions."

2. **The results in Figure 6d-f, which depict IWW for electricity and gas production and supply, are represented by scattered points. Can these scattered points reflect the general trend?**

**Response:** Thank you for the comments. We apologize for the confusion in understanding the Figure 6d-f which showed the actual spatial pattern of IWW for electricity and gas production and supply. These scattered points showed the actual presence of enterprises that consume water.

We revised the illustration of Figure 6 to avoid confusion:

"Figure 6: Zoomed view of IWW in the densely urbanized regions in China at a spatial resolution of 0.01°, including the Beijing-Tianjin-Hebei region (a, d), Yangtze River Delta (b, e), and Pearl River Delta (c, f). Panels (a)–(c) show the spatial pattern of IWW for manufacturing, and Panels (d)–(f) show the spatial pattern of IWW for electricity and gas production and supply. The numbers displayed as percentages denote the percentage of the sectoral IWW to total IWW."

**Therefore, the author should address these issues in greater detail and providing a more comprehensive discussion of the uncertainties and limitations of the study.**

**Response:** Thank you for the comments.

Based on the comments of all reviewers and the study's limitations, we strengthen the discussion of uncertainties present in the study regarding water use efficiencies in spatial downscaling, provincial differences of seasonal variations, and the feasibility of using the constant spatial pattern to allocate IWW for long periods.

(a) Water use efficiencies of enterprises of different sizes in spatial downscaling

In reality, the water use efficiency of a given enterprise could be different from other enterprises even for the same subsector, due to investment, technology, revenue, scale and so on. It is reasonable that enterprises of different sizes tend to have different water use efficiencies, and it is possible that larger companies may have higher water use efficiency than smaller ones. However, currently, we do not have data to provide specific information about the enterprise sizes and their water use efficiencies. If we arbitrarily introduce this scaling relationship without actual data, this would bring new uncertainty to spatial distribution. In the future, when such data becomes available, incorporating this information could better estimate enterprise-level IWW.

We modified this part to be more specific in the revision.

"This assumption disregarded the WUE variations since the WUE of different enterprises could vary substantially depending on subsector, technological levels, investment, scale effects and so on. For this matter, the spatial distribution of IWW could be further improved with better data sources at finer scales in the future."

(b) Provincial differences in seasonal variations

There are indeed regional differences in the seasonality of IWW. Figure R2 shows the monthly variations of production output across provinces for different industrial sectors, and we can see most of them follow some differences in different provinces (e.g., for electricity and manufacturing sectors). However, at the provincial level, the seasonal fluctuations may exhibit unreasonable or chaotic patterns that are hard to explain, such as manufacturing and mining sectors of Hainan, Guangxi province. For example, Tibet's fraction of manufacturing production in January and February was too low, under 0.025. The exact reasons are unclear,

but they could be caused by statistical/random errors in the data. Therefore, we used each subsector's national mean monthly variations to allocate IWW instead of provincial-level seasonal variations which are problematic in certain places. This choice would not affect much the seasonality of the final IWW because the seasonality of different sectors plays a dominant role in determining the seasonality of IWW for a province (Reynaud, 2003; Sathre et al., 2022).

[Figure]

**Figure R2 (Figure C3).** Seasonal variations in the national total IWW (a) and provincial IWW for separate industrial sectors, including the electricity and gas production and supply (EGPS) (b), manufacturing (c), and mining sectors (d). The seasonal variations are represented as the fraction of monthly IWW to the annual total during 2006-2010. The thick lines represent the water withdrawal of sectors, and the thin lines represent provinces. The shadows represent the seasons with peak and low water withdrawals in a year.

We added Figure R1 as Figure 3C and a more detailed discussion on seasonal variations among provinces in the revised manuscript.

"We observed considerable differences in monthly variations in production output across provinces for different industrial sectors (Fig. C3). However, the seasonal fluctuations shown in sectors, such as manufacturing and mining, exhibited patterns that were chaotic and unreasonable at the provincial level (Fig. C3). It was difficult to determine whether these different seasonal fluctuations originated from statistical/random errors, unweighted product outputs to the subsector, interannual variability, or actual regional differences. Therefore, we selected to use the national mean monthly variations to represent each subsector to improve the robustness. These monthly subsector variations were then combined with the subsectoral water withdrawal to derive the seasonal variations in IWW (Eq. 4). This choice was expected to have a limited impact on the seasonality of total IWW because it was primarily determined by the sector composition of a province (Reynaud, 2003; Sathre et al., 2022). In future research, the regional differences in seasonal variations in IWW should be further explored."

(c) The feasibility of using the constant spatial pattern to allocate IWW for long periods.

We used water use efficiency and the resulting spatial-seasonal patterns of IWW in 2008 to downscale IWW in other years from 1965 to 2020, serving as a time-invariant pattern for downscaling. First, we made this choice mainly because of the data constraint since no data were available to calculate subsector water use efficiency for years other than 2008. This practice is not ideal but is justifiable given the data limit and the practices adopted in other studies. Developing long-time series gridded data of IWW based on either a time-invariant pattern (e.g., H08, WaterGAP3, and PCR-GLOBAL) or patterns with decadal updates (e.g., Huang et al., 2018; Dong et al., 2022) for downscaling can be found in previous studies (Table R1). Second, industrial water use efficiency generally improved over time with the development of technology. This means that the temporal improvement in water use efficiency is likely to apply for all enterprises (Chen et al., 2019), while the spatial differences in water use efficiency of a given year are still determined by the spatial distribution of economic conditions which remain relatively stable over the years. Influence of long-term factors could be reflected through the changes in total IWW from statistical IWW data. For the above reasons, we chose the approach to develop the long-term gridded IWW data.

The dataset developed based on the time-invariant pattern of 2008 should be reasonably well for recent ~20 years, but it may contain larger biases for earlier years. We added this vital point to the manuscript to make the users aware of this issue so that they can choose the period of the data for their specific needs and accuracy considerations.

**Table R1 Spatial pattern used for long-term data extension in previous studies**

| Spatial pattern | Long-term data | Used for | Reference |
| --- | --- | --- | --- |
| NASA Back Marble night-time light intensity map in 2012-2016 | 1980-2016 | Model (VIC-5) | Droppers et al., 2020 |
| Distribution of urban population in 2009 | 1950-2010 | Model (WaterGAP3) | Flörke et al., 2013 |
| Global population distribution map and | 1970-2010 | Model (H08) | Hanasaki et al., |

| | | | |
|---|---|---|---|
| national boundary information in 2005 | | | 2008 |
| Global IWW map in 2000 | 1960-2001 | Model (PCR-GLOBWB) | Wada et al., 2011a, b |
| Linear Interpolation based on GDP dataset in 1990, 2000 and 2010, same as 1990 before 1990 | 1971-2010 | Model (CLHMS, the Coupled Land Surface-Hydrologic Model System) | Dong et al., 2022 |
| Global population density maps with decadal updates (1980, 1990, 1995, 2000, 2005) | 1970-2010 | Water Dataset | Huang et al., 2018 |

We added a more detailed discussion on this matter in the revision.

Firstly, we modified this part to be more specific:

"Nevertheless, this practice was acceptable in the literature under the data limit. For example, time-invariant spatial patterns (e.g., H08, WaterGAP3, and PCR-GLOBAL) or patterns with decadal updates (e.g., Huang et al., 2018) were used when developing the gridded IWW data with long time spans. Other time-varying data sources, such as nightlight, land cover, and population density maps with frequent temporal updates, could potentially facilitate the characterization of the temporal changes in the spatial pattern of IWW. "

Besides, we added a discussion on the influence of the industrial WUE and stated the stable distribution of population and economy of the country.

"The long-term changes in the industrial WUE can affect IWW, since WUE generally improves over time with the development of technology. This improvement would occur for all enterprises (Chen et al., 2019; Yang et al., 2021) and thus may not necessarily change the broad spatial pattern of IWW; this pattern is determined by the spatial distribution of industry and economic activities. The influence of other long-term factors could be captured by the changes in the total IWW from the statistical data."

"These analyses support the fact that specific industrial enterprises, their WUE, and water withdrawal substantially changes over time, and the broad spatial pattern after aggregating to the grid scale can still be applied because the spatial pattern of IWW is largely determined by the distribution of the population and economy of the country, which remain relatively stable over the years."

Additionally, we specified limitations of the CIWW dataset for users:

"Due to this limitation, the CIWW dataset would have better performance for the last 20 years but may contain larger uncertainties towards earlier periods. Users can select the time period of the dataset according to their specific needs and interpret earlier years data with caution."

**Reference**

Addor, N., Do, H. X., Alvarez-Garreton, C., Coxon, G., Fowler, K., and Mendoza, P. A.: Large-sample hydrology: recent progress, guidelines for new datasets and grand challenges, Hydrol. Sci. J., 65, 712–725, https://doi.org/10.1080/02626667.2019.1683182, 2020.

Alcamo, J., Flörke, M., and Märker, M.: Future long-term changes in global water resources driven by socio-economic and climatic changes, Hydrol. Sci. J., 52, 247–275, https://doi.org/10.1623/hysj.52.2.247, 2007.

Chen, Q., Ai, H., Zhang, Y., and Hou, J.: Marketization and water resource utilization efficiency in China, Sustain. Comput. Inform. Syst., 22, 32–43, https://doi.org/10.1016/j.suscom.2019.01.018, 2019.

Dong, N., Wei, J., Yang, M., Yan, D., Yang, C., Gao, H., Arnault, J., Laux, P., Zhang, X., Liu, Y., Niu, J., Wang, H., Wang, H., Kunstmann, H., and Yu, Z.: Model Estimates of China's Terrestrial Water Storage Variation Due To Reservoir Operation, Water Resour. Res., 58, https://doi.org/10.1029/2021WR031787, 2022.

Droppers, B., Franssen, W. H. P., Van Vliet, M. T. H., Nijssen, B., and Ludwig, F.: Simulating human impacts on global water resources using VIC-5, Geosci. Model Dev., 13, 5029–5052, https://doi.org/10.5194/gmd-13-5029-2020, 2020.

Flörke, M., Kynast, E., Bärlund, I., Eisner, S., Wimmer, F., and Alcamo, J.: Domestic and industrial water uses of the past 60 years as a mirror of socio-economic development: A global simulation study, Glob. Environ. Change, 23, 144–156, https://doi.org/10.1016/j.gloenvcha.2012.10.018, 2013.

Hanasaki, N., Kanae, S., Oki, T., Masuda, K., Motoya, K., Shirakawa, N., Shen, Y., and Tanaka, K.: An integrated model for the assessment of global water resources – Part 1: Model description and input meteorological forcing, Hydrol. Earth Syst. Sci., 12, 1007–1025, https://doi.org/10.5194/hess-12-1007-2008, 2008.

Hejazi, M., Edmonds, J., Clarke, L., Kyle, P., Davies, E., Chaturvedi, V., Wise, M., Patel, P., Eom, J., Calvin, K., Moss, R., and Kim, S.: Long-term global water projections using six socioeconomic scenarios in an integrated assessment modeling framework, Technol. Forecast. Soc. Change, 81, 205–226, https://doi.org/10.1016/j.techfore.2013.05.006, 2014.

Huang, Z., Hejazi, M., Li, X., Tang, Q., Vernon, C., Leng, G., Liu, Y., Döll, P., Eisner, S., Gerten, D., Hanasaki, N., and Wada, Y.: Reconstruction of global gridded monthly sectoral water withdrawals for 1971-2010 and analysis of their spatiotemporal patterns, Hydrol. Earth Syst. Sci., 22, 2117–2133, https://doi.org/10.5194/hess-22-2117-2018, 2018.

Koch, H. and Vögele, S.: Dynamic modeling of water demand, water availability and adaptation strategies for power plants to global change, Ecol. Econ., 68, 2031–2039, https://doi.org/10.1016/j.ecolecon.2009.02.015, 2009.

Liu, X., Liu, W., Yang, H., Tang, Q., Flörke, M., Masaki, Y., Müller Schmied, H., Ostberg, S., Pokhrel, Y., Satoh, Y., and Wada, Y.: Multimodel assessments of human and climate impacts on mean annual streamflow in China, Hydrol. Earth Syst. Sci., 23, 1245–1261, https://doi.org/10.5194/hess-23-1245-2019, 2019.

Long, D., Yang, W., Scanlon, B. R., Zhao, J., Liu, D., Burek, P., Pan, Y., You, L., and Wada, Y.: South-to-North Water Diversion stabilizing Beijing's groundwater levels, Nat. Commun., 11, 3665, https://doi.org/10.1038/s41467-020-17428-6, 2020.

Otaki, Y., Otaki, M., and Yamada, T.: Attempt to Establish an Industrial Water Consumption Distribution Model, J. Water Environ. Technol., 6, 85–91, https://doi.org/10.2965/jwet.2008.85, 2008.

Reynaud, A.: An Econometric Estimation of Industrial Water Demand in France, Environ.

Resour. Econ., 25, 213–232, https://doi.org/10.1023/A:1023992322236, 2003.

Sathre, R., Antharam, S. M., and Catena, M.: Water Security in South Asian Cities: A Review of Challenges and Opportunities, CivilEng, 3, 873–894, https://doi.org/10.3390/civileng3040050, 2022.

Wada, Y., Van Beek, L. P. H., Viviroli, D., Drr, H. H., Weingartner, R., and Bierkens, M. F. P.: Global monthly water stress: 2. Water demand and severity of water stress, Water Resour. Res., 47, 1–17, https://doi.org/10.1029/2010WR009792, 2011a.

Wada, Y., Beek, L. P. H. V., and Bierkens, M. F. P.: Modelling global water stress of the recent past: on the relative importance of trends in water demand and climate variability, Hydrol. Earth Syst. Sci., 15, 3785–3808, https://doi.org/10.5194/hess-15-3785-2011, 2011b.

Wada, Y., Flörke, M., Hanasaki, N., Eisner, S., Fischer, G., Tramberend, S., Satoh, Y., Van Vliet, M. T. H., Yillia, P., Ringler, C., Burek, P., and Wiberg, D.: Modeling global water use for the 21st century: The Water Futures and Solutions (WFaS) initiative and its approaches, Geosci. Model Dev., 9, 175–222, https://doi.org/10.5194/gmd-9-175-2016, 2016.

Zhou, F., Bo, Y., Ciais, P., Dumas, P., Tang, Q., Wang, X., Liu, J., Zheng, C., Polcher, J., Yin, Z., Guimberteau, M., Peng, S., Ottle, C., Zhao, X., Zhao, J., Tan, Q., Chen, L., Shen, H., Yang, H., Piao, S., Wang, H., and Wada, Y.: Deceleration of China's human water use and its key drivers, Proc. Natl. Acad. Sci. U. S. A., 117, 7702–7711, https://doi.org/10.1073/pnas.1909902117, 2020.

---

## Author Response (AR2)

**Response to Comments on the Manuscript (essd-2023-66):**

We deeply appreciate the detailed and constructive comments provided by the two anonymous reviewers during the second-round of review. Their suggestions and comments have been invaluable in refining our work, and we have carefully revised the manuscript and provided a point-to-point response to each comment.

The original comments are in **bold font**, our response is in regular font, and the changes in the text are in blue.

**Response to Comments of Referee #1:**

**This revised manuscript proposes a gridded dataset detailing industrial water withdrawal (IWW) in China. In contrast to prior datasets, the dataset spans from 1965 to 2020, incorporating a seasonal cycle and showcasing enhanced spatial resolutions. Specifically, the seasonal variability refers to a dataset in the period 2006-2010, the spatial pattern refers to data from 2008, and subsequently, the collected provincial industrial IWW survey records from two other studies, which cover 1965 to 2020, are reconstructed into gridded maps based on the spatial and seasonal variabilities.**

**Following the first-round revision, incorporating feedback from three reviewers, improvements have been made to the paper. However, aligning with the concerns raised by the reviewers, who comprehensively covered my questions, I intend to provide my thoughts to the response of the authors. The current explanations remain unclear to me. While I acknowledge the overall strength of the study's motivation, reliance on these less-convincing assumptions does not effectively address the issues outlined by the authors in the introduction.**

**Response:** Thank you very much for taking the time to review our manuscript and provide valuable feedback and comments. We appreciate your detailed thoughts on our assumptions on the spatialization and seasonality employed in developing industrial water withdrawal (IWW) data. Following your suggestions and comments, we added more supplement analysis to improve the confidence of the assumptions and revised texts to further clarify the assumptions and uncertainty of data.

**Major:**

**Concern one: input spatial pattern of WUE (water use efficiency) from 2008 to represent the whole period from 1965 to 2020:**

**To support this assumption, the authors have replied that some models also utilized in time-invariant spatial patterns, and "spatial pattern of IWW is largely determined by the distribution of the population and economy of the country, which remain relatively stable over the years." I think the stable economic pattern can be only assumed for developed countries from 1965 to 2020, but for China, the spatial pattern in the last 60 years has tremendously changed, and IWW also depends on industry styles that have been updated and transferred from province to province substantially.**

**Response:** Thank you for your comments and we would like to further clarify this important point.

The long-term gridded IWW data from 1965 to 2020 was developed by downscaling statistical IWW data based on the spatial-seasonal patterns of IWW in 2008. Although we could not directly incorporate the time-varying WUE at the grid level owing to the lack of such data, the provincial-level IWW statistical data have partially accounted for the changing spatial pattern of national IWW driven by economic development and changing WUE at the province level (Figure R1). For example, Shanghai's IWW was a significant proportion of the national total at an early time, but it declined sharply from 0.16 to 0.07 by the late 1970s. Similarly, the proportion of IWW in Heilongjiang and Liaoning gradually decreased between 1965 and 2020 (from 0.08 to 0.018 and 0.053 to 0.016 respectively). Conversely, the share of IWW in Jiangsu, Jiangxi, and Anhui exhibited a gradual increase between 1965 and 2020 (from 0.08 to 0.23, 0.004 to 0.05, and 0.02 to 0.08, respectively). The changing proportion of IWW of each province through time enables us to capture the changes in provincial-level spatial patterns. However, we acknowledge that this cannot track the changing spatial pattern of IWW within the province. We added Figure R1 to Figure I9 and modified the related sentence in the revision.

"The influence of other long-term factors such as climate change and WUE changes related to industry development could be partially captured by the provincial statistical data which incorporate the changing spatial pattern of total IWW at the provincial level (Fig. I9)."

[Figure]

Figure R1/I9. The changing proportion of provincial IWW to national total from 1965 to 2020

**Moreover, it is crucial to distinguish between "time-invariant input" and "input from a single year". In other words, even if the parameter is assumed to be time-invariant, the time-invariant pattern should be derived as the average across multiple years rather than from one single year.**

**In addition, I have double-checked the references the author cited, these models only mentioned the WUE increase but didn't explicitly give the reason why the time-invariant assumption is reasonable.**

**Response:** In fact, the spatial-seasonal pattern of IWW in China would be different with the industrial development. Ideally, downscaling industrial water withdrawals using yearly-varying spatial-temporal patterns is the most desirable option, or to a lesser extent by using the time-invariant pattern derived from an average of multiple years. However, the China Economic Census Yearbook in 2008 is the only data source currently available for the subsectoral IWW over the past 60 years. The lack of data in other years keeps us from using the yearly varying or average of spatial-temporal patterns across multiple years. Therefore, we can only use the spatial-season pattern in 2008 to downscale the long-term provincial IWW. This choice is not perfect but we feel it is acceptable given the data limitation. Besides, the examples mentioned in the discussion such as WaterGAP3 (Flörke et al., 2013) and PCR-GLOBAL (Wada et al., 2011a, b) used the distribution of urban population and IWW from the single year to downscale calculations of long-term IWW data, while H08 (Hanasaki et al., 2008) used 5-year distribution of population for calculating the long-term data. In these model data, increasing WUE was used to estimate changes in IWW globally, but not for the spatial downscaling of long-term data. In the revision, we revised the text to specify what "time-invariant" means.

"We acknowledge that the time-invariant spatial-seasonal pattern of IWW from a single year in 2008 was a strong assumption and probably not true in reality. Nevertheless, this practice was acceptable in the literature under the data limit. For example, the spatial patterns from a single year (e.g., the urban population distribution in 2009 used in WaterGAP3 and Global IWW map in 2000 used in PCR-GLOBAL) or patterns with multi-year updates (e.g., H08 and Huang et al., 2018) were both used when developing the gridded IWW data with long time spans."

**Table R1 Spatial pattern used to derive long-term data in previous studies**

| Spatial pattern | Long-term data | Used for | References |
|---|---|---|---|
| Distribution of urban population in 2009--Input from a single year | 1950-2010 | Model (WaterGAP3) | Flörke et al., 2013 |
| Global IWW map in 2000--Input from a single year | 1960-2001 | Model (PCR-GLOBWB) | Wada et al., 2011a, b |
| Global population distribution map (1990, 1995, 2000) and national boundary information in 2005 | 1970-2010 | Model (H08) | Hanasaki et al., 2008 |
| Global population density maps with decadal updates (1980, 1990, 1995, 2000, 2005) | 1970-2010 | Global gridded monthly sectoral water use dataset | Huang et al., 2018 |

**Additionally, the authors mentioned the unavailability of the data in other years, but why Yearbooks of China in other years did not provide such statistics?**

**Response:** Thank you for giving us the opportunity to clarify this point.

The provincial-level subsectoral IWW data by 36 subsectors is from economic censuses (the China Economic Census Yearbook in 2008). Although China has conducted 5 economic censuses since 2004 (2004, 2008, 2013, 2018, and 2023 have not yet been released), only the 2008 China Economic Census data included national and provincial IWW by subsector. Besides, other statistical data sources did not have the IWW data with detailed subsectoral information as those in the 2008 China Economic Census data. Therefore, we only used 2008 data for our estimation. We modified the related sentence in the revision.

"To match the IWW survey data, which were only available in 2008 (the economic censuses in other years do not include detailed provincial IWW by subsector), industrial enterprise data in 2008 were selected for spatial downscaling of the provincial IWW (Fig. B2)."

**Last but not least, the authors have provided discussion to state the high uncertainty of the proposed data in early years due to such assumption, but it would be helpful to give some quantitative analysis to prove how the uncertainty changed caused by inputting WUE pattern only from 2008.**

**Response:** Following the reviewer's suggestion, we compared the three gridded IWW datasets (CIWW, Huang data, and model data) against Zhou2020 data at the prefecture level (whose information was not used in developing CIWW) to evaluate their uncertainty in earlier years. Results in Figure R2 show that CIWW clearly outperforms Huang and model data, with much higher correlation and smaller RRMSE (Relative Root Mean Squared Error, RMSE/mean) compared with Zhou2020 data, especially in the early years. This indicates that the performance of CIWW is still the best among the three despite the uncertainty. In the revised manuscript we added Figure R2 to Figure K11 and modified the related sentence:

"Users can select the time period of the dataset according to their specific needs and interpret earlier years' data with caution. Nevertheless, the CIWW data in earlier years showed surprisingly good performance with a much higher correlation (0.80 vs. 0.39~0.43 in 1971; as illustrated in Fig. K11) and smaller RRMSE (Relative Root Mean Squared Error, RMSE/mean, 1.81 vs. 2.67~ 2.76 in 1971) than other datasets when compared against Zhou2020 data at prefectural level (Note the prefecture-level IWW from Zhou2020 data was not used in developing CIWW)."

[Figure]

Figure R2/K11. Comparing the three gridded IWW (CIWW, Huang data, and model data) against Zhou2020 data at the prefecture-level from 1965 to 2010. Higher is better for correlation, and lower is better for RRMSE(Relative Root Mean Squared Error, RMSE/mean).

**Concern two: The validation data in Figure 2 is not independent from the input data,**

despite their differing scales. I recommend incorporating additional physically relevant variables for comparison with the proposed CIWW. This comparative analysis, both spatially and temporally, would demonstrate the new dataset's responsiveness to independent data sources and its potential utility for scientific analysis.

**Response:** Thank you for the comments.

It would be great to have other independent IWW data or physically relevant variables for validation. Unfortunately, such data might still be lacking and we could not find usable data after searching. We look forward to making more comparisons if relevant IWW data come out in the future. Given the data constraint, we deliberately used the provincial-level IWW of Zhou2020 data to develop the CIWW data while leaving the prefectural-level data only for validation. Although prefectural-level and provincial-level IWW are not considered to be completely independent (each province consists of many prefectures), the spatial variations in prefecture-level IWW at a finer scale are not captured by the provincial IWW. In the absence of additional data, this prefectural IWW can support the validation of existing datasets in China. Following this suggestion, we revised the sentence in section 2.3:

"Although we used the Zhou2020 data at the provincial level to produce the CIWW dataset, the prefectural-level data were unused in developing CIWW but left intentionally only for validation purposes. The provincial- and prefectural-level IWW are not completely independent (each province consists of many prefectures), however, the intra-provincial variations reflected in prefectural IWW are not captured by the provincial IWW. In the absence of additional validation data, the prefectural IWW can support the validation and determine the effectiveness of spatial downscaling."

**Concern three: The weaker seasonal pattern observed in the proposed data's seasonal cycle (depicted by the blue line in Figure 2b) is shown, which is mainly caused by overestimated values in winter. Could you please provide further details or insights into the reasons behind this observation?**

**Response:** Thank you for giving us the opportunity to clarify this point.

Since seasonal variations of CIWW are derived from the monthly fraction of national subsectoral industries and IWW, reflecting the seasonality at the national level (except Electricity and Heating Power Production and Supply subsector which used regional different seasonality in the revised manuscript). The national seasonal variation applied in Beijing may not fully capture local variations. The overestimated IWW in December could be caused by a combination of the high heating energy demand and the increased output of manufacturing stimulated by international trade. We added this supplement in the 2.3 section.

"The results showed that both the CIWW and Huang data could capture the 5-year mean seasonality of IWW in Beijing (Fig. 2b). The slight deviation of CIWW from statistical data in certain months (e.g., December) reflects the imperfect capability of applying national seasonality to characterize local variations in Beijing."

**Concern four: Figure 5, any explanation for the IWW decrease at a country level after 2010?**

**Response:** Thank you for giving us the opportunity to clarify this point.

The decline of national IWW after 2010 is mainly due to the implementation of a series of

water-saving management measures (The State Council of the People's Republic of China, 2011) such as establishing "three red lines" to cap the total water withdrawal, enhance water use efficiency (WUE), and restrict pollutants in water function areas. As a result, the industrial recycling water rate exhibited a gradual increase over time from 85.7% in 2010 to 92.5% in 2020 (Chen and Chen, 2021), while the industrial water use intensity gradually decreased from 90 in 2010 to 32.9 in 2020 (unit: $m^3$/10000 yuan) (Zhang et al., 2023).

To explain the decline in national IWW, we added an implementation in the revision.

"These long-term changes indicated that IWW in China has now entered a slowly declining phase. The decline of national IWW after 2010 is mainly due to the implementation of a series of water-saving management measures (The State Council of the People's Republic of China, 2011) such as establishing "three red lines" to cap the total water withdrawal, enhance water use efficiency, and increase industrial water recycling rate (Chen and Chen, 2021; Zhang et al., 2023)."

**There is input replacement before and after 2002, whereas the result in Fig 5 shows a perfectly smooth temporal variation, any process regarding this input replacement? Only because the two input datasets well match each other?**

The statistical data of IWW consists of two sources: Zhou2020 data for 1965-2013 and China Water Resources Bulletin for 2003 afterward. During the overlap periods, the national IWW from the two data sources was almost identical (117.72 vs 118.86 unit: $km^3$) in 2003 (Figure 3R) but started to diverge for the rest years. We opted for the China Water Resources Bulletin starting from 2003 as the data source because it is being updated continuously. Therefore, the monthly national IWW exhibits a smooth curve, as shown in Figure 3R. In the revised manuscript we added Figure R3 as Figure C3 and a more detailed description in the section of data as follows:

"The national IWW between two sources (Zhou2020 data and China Water Resources Bulletin) was almost identical in 2003 (117.72 vs 118.86 unit: $km^3$; Fig. C3) but started to diverge afterward. To ensure data continuity, we opted for the China Water Resources Bulletin starting from 2003 as a statistical data source because it has been updated continuously since then. Thus, the combination of the above two data sources provided complete and continuous statistical records of IWW from 1965 to 2020 in China."

[Figure]

Figure R3/C3. The national IWW from two statistical data sources.

**As you already aggregated the data variation analysis to country and decade scales, what are other IWW data performances in this analysis? A comparison is still needed.**

Following the reviewer's suggestion, we plotted the long-term monthly/annual national IWW of CIWW, Huang and model data (monthly data unavailable) together (Figure 4R). The comparison indicates that the other two datasets overall underestimated China's total IWW. We also added the comparison results in the revision.

"In addition, the comparison of long-term annual national IWW of three datasets (CIWW, Huang and model data) showed that the other two datasets significantly underestimated China's total IWW and presented different temporal patterns, which could not consider the effects of water use policies (Fig. H8).

[Figure]

Figure 4R/H8 Monthly and yearly IWW in China from 1965 to 2020 from the CIWW, Huang and model data.

**Minor:**

**(1) Line 24: I understand you would like to point out the definition distinction between IWW and water consumption, but this sentence reads like "IWW in this study doesn't include water consumption part".**

**Response:** Thanks for kindly reminding us of this issue.

We modified this sentence and the revised texts are shown below:

"Industrial water withdrawal (IWW) is the amount of water abstracted from fresh water sources for industrial purposes, which is different from water consumption;"

**(2) Figure 6 d-f: it is really difficult to find scatters and corresponding colors on these three subplots.**

**Response:** Thank you for the comments. We revised panel d-f of this figure to show results at 0.02° resolution to improve readability. The revised Figure 6 is as follows:

[Figure]

Figure 5R/6. Zoomed view of IWW in the densely urbanized regions in China at a spatial resolution of 0.01° (a, b, c) and 0.02° (d, e, f) for clarity, including the Beijing-Tianjin-Hebei region (a, d), Yangtze River Delta (b, e), and Pearl River Delta (c, f). Panels (a)–(c) show the spatial pattern of IWW for manufacturing, and Panels (d)–(f) show the spatial pattern of IWW for electricity and gas production and supply. The numbers displayed as percentages denote the percentage of the sectoral IWW to total IWW.

**Response to Comments of Referee #2:**

**This manuscript developed a gridded dataset of monthly industrial water withdrawal in China from 1965 to 2020, with a spatial resolution of 0.1°and 0.25°. While the temporal and spatial coverage of the dataset is extensive, I have some minor reservations regarding the spatialization method employed in the study. I'd suggest to address these issues before acceptance for publication.**

**Response:** We would like to sincerely express our gratitude to you for your careful reading and constructive comments. According to the comments, we have tried our best to improve the manuscript, and a point-by-point response follows.

**1. My biggest concern is that the spatial distribution of industrial water use over a long time series is based on enterprise data from 2008. However, as economic development progresses and regional development trends, the spatial distribution of both enterprises and industrial water withdrawal is likely to change. These changes can potentially reduce the spatial accuracy of the long-term industrial water withdrawal data. Besides that, as you mentioned in the discussion section, the underlying assumption of this study is 'This improvement would occur for all enterprises', which means the water use efficiency of all enterprises has improved simultaneously. However, this assumption is highly uncertain and may not reflect the actual situation. Would there be other possible data sources you**

**can use to improve the situation? I'd highly recommend improving this.**

Response: Thank you for your comment and your concern is legit.

Yes, using the spatial-seasonal pattern of 2008 to downscale multi-year IWW may affect the spatial accuracy of the long-term CIWW. We primarily made this decision due to data constraints, as there was no available data to estimate subsector or enterprises' water use efficiency for years other than 2008. Although we could not directly incorporate the time-varying WUE at the grid level, the provincial-level IWW statistical data have accounted for the changing spatial distribution of national IWW at the province level driven by economic development and changing WUE (Figure R1). For example, the proportion of IWW in Heilongjiang and Liaoning gradually decreased between 1965 and 2020 (from 0.08 to 0.018 and 0.053 to 0.016 respectively). Conversely, the share of IWW in Jiangsu, Jiangxi, and Anhui exhibited a gradual increase between 1965 and 2020 (from 0.08 to 0.23, 0.004 to 0.05, and 0.02 to 0.08, respectively). However, we acknowledge that this cannot track the changing spatial pattern of IWW within the province.

To evaluate the spatial accuracy in earlier years, we compared three gridded IWW datasets (CIWW, Huang data, and model data) against Zhou2020 data at the prefecture level (whose information was not used in developing CIWW) to evaluate their effectiveness in characterizing the spatial pattern of IWW. Results in Figure R2 show that CIWW clearly outperforms Huang and model data, with much higher correlation and smaller RRMSE (Relative Root Mean Squared Error, RMSE/mean) compared with Zhou2020 data, especially in the early years. This indicates that the performance of CIWW is still the best despite the uncertainty of water use efficiency.

In the revision, we added Figure R1 to Figure I9, Figure R2 to Figure K11, with further additional explanation in the discussion section.

"The influence of other long-term factors such as climate change and WUE changes related to industry development could be partially captured by the provincial statistical data which incorporate the changing spatial pattern of total IWW at the provincial level (Fig. I9)."

"Users can select the time period of the dataset according to their specific needs and interpret earlier years' data with caution. Nevertheless, the CIWW data in earlier years showed surprisingly good performance with a much higher correlation (0.80 vs. 0.39~0.43 in 1971, as illustrated in Fig. K11) and smaller RRMSE (Relative Root Mean Squared Error, RMSE/mean, 1.81 vs. 2.67~ 2.76 in 1971.) than other datasets when compared against Zhou2020 data at prefecture level (Note the prefecture-level IWW from Zhou2020 data was not used in developing CIWW)."

[Figure]

Figure R1/I9. The changing proportion of provincial IWW to the national total from 1965 to 2020

[Figure]

Figure R2/K11. Comparing the three gridded IWW (CIWW, Huang data, and model data) against Zhou2020 data at the prefecture-level from 1965 to 2010. Higher is better for correlation, and lower is better for RRMSE(Relative Root Mean Squared Error, RMSE/mean).

**2. The China Water Resources Bulletin is available starting from 1997. However, in this study, only statistical data from 2003 to 2020 were used and merged with Zhou2020 data from 1965 to 2002. It is important to understand the rationale behind this decision. Regarding the merging of data with Zhou2020 dataset, it is essential to provide a relatively detailed explanation of Zhou2020 data and the methodology used. This would enable readers to evaluate the reliability and relevance of the data.**

**Response:** Thanks for kindly reminding us of this issue.

China's Water Resources Bulletin has been published since 1997, however, the provincial IWW data needed by this study only became available continuously after 2003 (it was available in a

few years such as 1999 and 2000 before 2003).

The Zhou2020 data (Zhou et al., 2020) integrate multiple versions of water resources survey data (1st and 2nd National Water Resources Assessment Program) and provide provincial/prefectural IWW from 1965 to 2013. During the overlap periods, the national IWW from the two data sources was almost identical (117.72 vs 118.86 unit: km$^3$) in 2003 (Figure R3) but started to diverge for the rest years. We opted for the China Water Resources Bulletin starting from 2003 as the data source because it is being updated continuously. Therefore, the monthly national IWW exhibits a smooth curve, as shown in Figure 3R.

In the revised manuscript we added Figure R3 as Figure C3 and a more detailed description in the section of data as follows:

"To further extend the time series to an earlier period, the IWW reported by Zhou et al., (2020) (referred to as 'Zhou2020 data' hereafter) from 1965 to 2002, was used after summing the prefecture data to the provincial level; its IWW was from multiple versions of water resources survey data (1st and 2nd National Water Resources Assessment Program) and defined the same way as the China Water Resource Bulletin and our study. The national IWW between two sources (Zhou2020 data and China Water Resources Bulletin) was almost identical in 2003 (117.72 vs 118.86 unit: km3; Fig. C3) but started to diverge afterward. To ensure data continuity, we opted for the China Water Resources Bulletin starting from 2003 as a statistical data source because it has been updated continuously since then. Thus, the combination of the above two data sources provided complete and continuous statistical records of IWW from 1965 to 2020 in China."

[Figure]

Figure R3/C3. The national IWW from two statistical data sources

**3. As mentioned in the Introduction, the interannual variations of IWW due to seasonal variations and climate fluctuations are crucial. However, the methodology employed in your study, which utilizes a uniform monthly ratio for the entire country, might not adequately represent the seasonal variations among provinces, given the vast expanse of China. While the discussion section notes that seasonal differences between sectors are more significant than those between provinces, there can still be significant seasonal variations within sectors due to factors such as climate patterns and economic development. This is particularly relevant for the electricity and gas production and supply sector, which is a major consumer of industrial water and highly sensitive to water temperature and technologies. A nationwide uniform approach may introduce significant uncertainties in estimating monthly water withdrawal.**

**Response:** Thanks for your constructive comments on this important point.

We agree that the seasonality of subsectors like Electricity and Heating Power Production and Supply is strongly linked to temperature seasonal variations of each province and thus may exhibit regional differences. Following the reviewer's suggestion, we further explored the different seasonality of IWW in the Electricity and Heating Power Production and Supply across provinces in China. We used the K-means method to cluster the seasonal variations into different spatial groups. The results showed that the seasonality in this subsector can be classified into three types, broadly corresponding to North China, South and Northwest China, and Xizang, respectively (Figure R4). The type 1 (15 provinces) seasonality has apparent two peaks around June and December, while type 2 peaked in JJA. Type 3 only includes Xizang whose seasonality looked different from other types. Shanghai, originally classified as type 1, was manually assigned to type 2 because of its strong peak in JJA.

[Figure]

Figure R4/F5. Classification of seasonal variations of provincial water withdrawal of Electricity and Heating Power Production and Supply to type 1 (a), type 2 (b), type 3 (c), and d. map of three types. The seasonal variations are the faction of monthly IWW to the annual total during 2006-2010. The thick color lines show the mean seasonally of the cluster, and the thin color lines show the seasonally of each province.

Based on the clustering analysis, we recreated the CIWW data based on the updated seasonal variation of three types (North China, South and Southwest China, and Tibet) of the subsector——Electricity and Heating Power Production and Supply. Besides, we re-compared the CIWW data with updated seasonality against the statistical data for seasonal variation in Beijing

(Figure 4R) and the difference was subtle.

[Figure]

Figure R5/2b. Comparison of the 5-year mean (2006-2010) monthly variation in IWW from the surveyed statistical data (red, (Long et al., 2020)), update CIWW (blue), raw CIWW (pink), and Huang data (green) in Beijing. The solid grey line shows IWW for individual years from 2006 to 2010. The inset shows the annual mean total IWW from 2006 to 2010.

In the revised manuscript we updated Figure R5 as Figure 2b, added Figure R4 as Figure F6, and a more detailed description in the 2.2.2 section as follows:

"Instead of directly using the provincial-specific seasonal variations of the output data, the seasonal variations in each industrial subsector ($\text{Fraction}_{mon,subs}^{water}$) were represented by the weighted mean of monthly product fractions across all provinces ($\text{Fraction}_{mon,p,subs}^{output}$) with weights of provincial subsector IWW ($IWW_{p,subs}$) from the Chinese Economic Census Yearbook in 2008 (Eq. 3). The only exception is for Electricity and Heating Power Production and Supply (EPS) subsector because its seasonality is strongly linked to seasonal temperature variation of each province and thus may exhibit regional differences. To account for this issue, we used the K-means method and classified the seasonality of EPS into three types, which broadly correspond to North China (type 1), South and Northwest China (type 2), and Xizang (type 3), respectively (Fig. F6). In particular, Shanghai was manually adjusted from the originally classified type 1 to type 2 because of its strong peak in JJA."

**4. Line203-205. The three models mentioned here may have different definitions for IWW. For instance, Water GAP considers the water withdrawal for thermoelectric power generation, while PCR-GLOBWB may not include such withdrawals. It is essential for the authors to clarify what type of data they are using from these models and whether their IWW definition aligns with the definitions used in the CIWW.**

Response: Thanks for kindly reminding us of this issue.

The model data used for comparison is Input Data used in ISIMIP2b (https://www.isimip.org/gettingstarted/details/38/, last accessed: 18 March 2024), which is water abstraction for industrial uses of the multi-model mean (from Water GAP, PCR-GLOBWB, and H08) from 1901 to 2005. The definition of IWW in the three models is not the

same. IWW in both H08 (manufacturing use and energy production) and PCR-GLPBWB means total IWW. Water GAP estimates manufacturing and energy water withdrawal separately and sums them to total IWW in the model simulation (Wada et al., 2016). All IWW in the model data does not include water withdrawals from the mining industry, but the impact would be small because mining only accounts for 5% of industrial water withdrawals according to statistical data.

To provide a clearer understanding of the data used for comparison, we have added a data introduction for comparison in the Data and Methods section.

"2.1.5 Other industrial water withdrawal data for comparison

There are two other gridded IWW datasets used to compare with the CIWW dataset: the global gridded monthly sectoral water use dataset for 1971-2010 at 0.5° (Huang et al., 2018) (referred to as 'Huang data' hereafter) and water abstraction for industrial uses from 1901 to 2005 at 0.5° as the input data for ISIMIP2b (referred to as 'model data' hereafter). The IWW from Huang data consists of three sectors: mining, manufacturing, and cooling of thermal power plants, and the sum of the three sectors was treated as the total IWW. The IWW from model data is the multi-model mean (Water GAP, PCR-GLOBWB, and H08). The sum of sectoral IWW (if available) was treated as total IWW. Unit of IWW was converted from $m^3$ to mm by dividing the grid cell area. Table D4 provides a summary of the data description used for comparison."

Table R2/D4 A summary description of other IWW data for comparison.

| Data variable | Data source | Industrial sector | Time span | Spatial resolution |
|---|---|---|---|---|
| Industrial water withdrawal | Global gridded monthly sectoral water use dataset | Sectors (3) | Monthly, 1971-2010 | 0.5° |
| Water abstraction for industrial uses | Input Data used in ISIMIP2b | None | Yearly, 1901-2005 | 0.5° |
| **Introduction of IWW between different models in model data** | | | | |
| **IWW in model** | **Industrial sector** | **Definition of IWW** | | |
| Water GAP | Sectors (2, except mining) | Total IWW is the sum of manufacturing and energy production water withdrawal | | |
| H08 | None | Total IWW includes manufacturing use and energy production. | | |
| PCR-GLOBWB | None | Total IWW no details available | | |

**5. Line215-217. From Fig2(b), it appears that there are significant discrepancies between the monthly variation line of CIWW and the survey statistics data for Beijing. The author only demonstrates consistency with the survey statistics data by comparing the total annual quantities, without explicitly addressing the consistency in seasonal variations.**

**Response:** Thank you for giving us the opportunity to clarify this point.

Seasonal variations of CIWW are derived from the monthly fraction of national subsectoral industries and IWW, reflecting the seasonality at the national level(except Electricity and Heating Power Production and Supply subsector which used regional different seasonality in the revised manuscript). The national seasonal variation applied in Beijing may not fully capture local variations. The overestimated IWW in December could be caused by a

combination of the high heating energy demand and the increased output of manufacturing stimulated by international trade. We added this supplement in the 2.3 section.

"The results showed that both the CIWW and Huang data could capture the 5-year mean seasonality of IWW in Beijing (Fig. 2b). The slight deviation of CIWW from statistical data in certain months (e.g., December) reflects the imperfect capability of applying national seasonality to characterize local variations in Beijing.

**Additionally, relying solely on data from one city to validate the entire dataset may not be appropriate. It is advisable to select multiple representative cities to provide evidence for the reliability of the data.**

**Response:** It would be great to have data in other cities to further validate seasonal variation of IWW, but unfortunately, we did not have the monthly IWW for other cities. We look forward to making more comparisons if relevant monthly IWW data come out in the future.

**6. Line236-237. It may be necessary to supplement the analysis with a population distribution map or other relevant data to demonstrate the linkage between manufacturing water withdrawal and population. Additionally, Fig3(c) presents results based on 2008 data, and the authors must provide evidence to support the continuity of this conclusion over a long time period.**

**Response:** Thank you for the constructive comments on this point.

Following the reviewer's suggestion, we displayed the population density maps of China every 5 years from 1990-2019 (Figure R5) (https://www.resdc.cn/doi/doi.aspx?doiid=32, last accessed: 18 March 2024), whose spatial distribution shows obvious similarity with the spatial distribution of manufacturing industry. Besides, given the development of the economy, the spatial pattern of population distribution remained stably over the nearly 30-year period from 1990 to 2019, indicating that the spatial pattern of manufacturing may remain relatively steady. In the revised manuscript we added Figure R5 as Figure G7.

"Water withdrawal from manufacturing broadly reflected the total IWW and population distribution of China, mainly showing the close linkage between manufacturing and population (Fig. 3c and Fig. G7)."

"These analyses support the fact that specific industrial enterprises, their WUE, and water withdrawal substantially changed over time, and the broad spatial pattern after aggregating to the grid scale can still be applied because the spatial pattern of IWW is largely determined by the distribution of the population and economy of the country, which remain relatively stable over the years (Fig. G7)."

[Figure]

Figure 5R/G7. The distribution pattern of population density in China from 1990 to 2019 (a-g) and Spearman's rank correlation coefficients of population density spatial pattern from 1990 to 2019 (h).

**7. In Fig3(b) and Fig6(d-f), the points on the graphs are not very clear, and it may be necessary to adjust the colorbar to make the results more distinct.**

**Response:** Thank you for the comments. We revised panel d-f of this figure to show results at 0.02° resolution to improve readability. The revised Figure 6 is as follows:

[Figure]

Figure 5R/6. Zoomed view of IWW in the densely urbanized regions in China at a spatial resolution of 0.01° (a, b, c) and 0.02° (d, e, f) for clarity, including the Beijing-Tianjin-Hebei region (a, d), Yangtze River Delta (b, e), and Pearl River Delta (c, f). Panels (a)–(c) show the spatial pattern of IWW for manufacturing, and Panels (d)–(f) show the spatial pattern of IWW for electricity and gas production and supply. The numbers displayed as percentages denote the percentage of the sectoral IWW to total IWW.

We uploaded a high-resolution version of Fig. 3(b) in the hope that it conveys the spatial information.

[Figure]

**Reference**

Chen, M. and Chen, H.: Spatiotemporal coupling measurement of industrial wastewater discharge and industrial economy in China, Environ. Sci. Pollut. Res., 28, 46319–46333, https://doi.org/10.1007/s11356-021-14743-3, 2021.

Flörke, M., Kynast, E., Bärlund, I., Eisner, S., Wimmer, F., and Alcamo, J.: Domestic and industrial water uses of the past 60 years as a mirror of socio-economic development: A global simulation study, Glob. Environ. Change, 23, 144–156, https://doi.org/10.1016/j.gloenvcha.2012.10.018, 2013.

Hanasaki, N., Kanae, S., Oki, T., Masuda, K., Motoya, K., Shirakawa, N., Shen, Y., and Tanaka, K.: An integrated model for the assessment of global water resources - Part 2: Applications and assessments, Hydrol. Earth Syst. Sci., 12, 1027–1037, https://doi.org/10.5194/hess-12-1027-2008, 2008.

Huang, Z., Hejazi, M., Li, X., Tang, Q., Vernon, C., Leng, G., Liu, Y., Döll, P., Eisner, S., Gerten, D., Hanasaki, N., and Wada, Y.: Reconstruction of global gridded monthly sectoral water withdrawals for 1971-2010 and analysis of their spatiotemporal patterns, Hydrol. Earth Syst. Sci., 22, 2117–2133, https://doi.org/10.5194/hess-22-2117-2018, 2018.

The state Council of the People's Republic of China, Decision on accelerating the reform and development of water resources (2011). https://www.gov.cn/gongbao/content/2011/content_1803158.htm(in Chinese)

Wada, Y., Van Beek, L. P. H., Viviroli, D., Drr, H. H., Weingartner, R., and Bierkens, M. F. P.: Global monthly water stress: 2. Water demand and severity of water stress, Water Resour. Res., 47, 1–17, https://doi.org/10.1029/2010WR009792, 2011a.

Wada, Y., Beek, L. P. H. V., and Bierkens, M. F. P.: Modelling global water stress of the recent past : on the relative importance of trends in water demand and climate variability, Hydrol. Earth Syst. Sci., 15, 3785–3808, https://doi.org/10.5194/hess-15-3785-2011, 2011b.

Wada, Y., Flörke, M., Hanasaki, N., Eisner, S., Fischer, G., Tramberend, S., Satoh, Y., Van Vliet, M. T. H., Yillia, P., Ringler, C., Burek, P., and Wiberg, D.: Modeling global water use for the 21st century: The Water Futures and Solutions (WFaS) initiative and its approaches, Geosci. Model Dev., 9, 175–222, https://doi.org/10.5194/gmd-9-175-2016, 2016.

Xu, X.: China population spatial distribution kilometer grid dataset. National Earth System Science Data Center, National Science & Technology Infrastructure of China. https://doi.org/10.12041/geodata.23724041532515.ver1.db

Zhang, L., Ma, Q., Zhao, Y., Chen, H., Hu, Y., and Ma, H.: China's strictest water policy: Reversing water use trends and alleviating water stress, J. Environ. Manage., 345, 118867, https://doi.org/10.1016/j.jenvman.2023.118867, 2023.

Zhou, F., Bo, Y., Ciais, P., Dumas, P., Tang, Q., Wang, X., Liu, J., Zheng, C., Polcher, J., Yin, Z., Guimberteau, M., Peng, S., Ottle, C., Zhao, X., Zhao, J., Tan, Q., Chen, L., Shen, H., Yang, H., Piao, S., Wang, H., and Wada, Y.: Deceleration of China's human water use and its key drivers, Proc. Natl. Acad. Sci. U. S. A., 117, 7702–7711, https://doi.org/10.1073/pnas.1909902117, 2020.